# Using rare genetic mutations to revisit structural brain asymmetry

Jakub Kopal [1,2], Kuldeep Kumar [3], Kimia Shafighi[1,2], Karin Saltoun [1,2], Claudia Modenato [4], Clara A. Moreau[5], Guillaume Huguet [3], Martineau Jean-Louis[3], Charles-Olivier Martin[3], Zohra Saci [3], Nadine Younis[3], Elise Douard [3], Khadije Jizi[3], Alexis Beauchamp-Chatel[6,7], Leila Kushan [8], Ana I. Silva [9,10], Marianne B. M. van den Bree [10,11,12], David E. J. Linden[9,10,12], Michael J. Owen [10,11], Jeremy Hall [10,11], Sarah Lippé[3], Bogdan Draganski [4,13], Ida E. Sønderby [14,15,16], Ole A. Andreassen [14,16], David C. Glahn[17], Paul M. Thompson[5], Carrie E. Bearden [8], Robert Zatorre [18,20], Sébastien Jacquemont[3,19] & Danilo Bzdok [1,2,20] ✉

Asymmetry between the left and right hemisphere is a key feature of brain organization. Hemispheric functional specialization underlies some of the most advanced human-defining cognitive operations, such as articulated language, perspective taking, or rapid detection of facial cues. Yet, genetic investigations into brain asymmetry have mostly relied on common variants, which typically exert small effects on brain-related phenotypes. Here, we leverage rare genomic deletions and duplications to study how genetic alterations reverberate in human brain and behavior. We designed a pattern-learning approach to dissect the impact of eight high-effect-size copy number variations (CNVs) on brain asymmetry in a multi-site cohort of 552 CNV carriers and 290 non-carriers. Isolated multivariate brain asymmetry patterns spotlighted regions typically thought to subserve lateralized functions, including language, hearing, as well as visual, face and word recognition. Planum temporale asymmetry emerged as especially susceptible to deletions and duplications of specific gene sets. Targeted analysis of common variants through genome-wide association study (GWAS) consolidated partly diverging genetic influences on the right versus left planum temporale structure. In conclusion, our gene-brain-behavior data fusion highlights the consequences of genetically controlled brain lateralization on uniquely human cognitive capacities.

No human brain is perfectly symmetrical, in contrast to what external body appearance may suggest. Architectural and functional differences between the left and right hemispheres are a fundamental design principle of brain organization[1,2]. Evolutionary, developmental, hereditary, experiential, and also pathological factors gave rise to lateralized specialization for some of the most developed capacities in humans[3]. As a prime example, functional brain asymmetries represent a core element of language that is uniquely evolved in humans[4,5]. In fact, a range of other cognitive functions also display degrees of hemispheric functional specialization, including aspects of motor skills, visuospatial and face processing, perception, learning, reasoning, and handedness[6]. It has been suggested that the organism's brain lateralization increases fitness via avoidance of unnecessary duplication of neuronal activity in both hemispheres, faster neuronal

processing due to not being constrained by slow callosal transfer of information between the hemispheres, and better coordination of unilateral behaviors[7]. Consequently, optimal brain functioning relies on a careful balance between the hemispheres where excess or lack of asymmetry may indicate a derailment in processes that influence hemispheric lateralization[8]. That is why aberrant asymmetries have been associated with numerous brain disorders[9], such as dyslexia[10], Alzheimer's disease[11], attention-deficit/hyperactivity disorder (ADHD)[12], autism spectrum disorder (ASD)[13], and psychotic disorders[14] including schizophrenia[15], but not major depression[16]. Although brain asymmetry presents a promising intermediate phenotype between behavior and disease, the genetic underpinnings underlying asymmetry remain largely unknown[6].

As early as the tenth week after conception, the majority of human fetuses display forms of behavioral asymmetries, such as preferential right, as opposed to the left, arm movement[17]. Neuroanatomical studies of fetuses and newborns[18,19], as well as embryonic gene expression analyses[20–22], demonstrated the prenatal emergence of brain asymmetry features. Later, the functional and anatomical differences are further accentuated during infancy as the language-related regions in the left hemisphere develop more slowly than their right counterparts[23,24]. In adulthood, over 90% of cortical brain regions display substantial left-right asymmetries in surface area measures, with the greatest asymmetries being localized to the perisylvian language area[6,25]. As such, hemispheric asymmetry is a fundamental organizational principle of the human brain, which is under the control of under-investigated genetic developmental programs[26].

The broadly lateralized programs of brain development, starting already *in utero*, are known to have a notable heritable component[27], estimated to be up to ~25%[6,28]. However, prior studies of genetic influences on the human brain's left-right axis have been dominated by common genetic variants – single nucleotide polymorphisms (SNPs) occurring in >=1% of the human population[28]. These variants, individually or in combination, are typically associated with only small explanatory effects on brain-related phenotypes[29]. Our understanding of the biology behind how SNPs impact phenotypes remains limited. This is due to i) challenges in ascertaining the causal contribution of common genetic variants identified by genome-wide association studies (GWAS), ii) difficulties linking regulatory elements to corresponding regulated protein-coding genes, and iii) the fact that the same common variant can have different effects in different body tissues[30]. In contrast to common SNPs, copy number variations (CNVs) have large and predictable effects on the level of expression of genes fully encompassed by these gene dosage alterations. CNVs are defined as deletions or duplications of continuous sequences of nucleotides more than 1000 base pairs long[31,32]. Advances in genomic microarray technology streamlined the detection of pathogenic CNVs that are as rare as 1 in 3600 for 22q11.2 deletion[33]. Hence, the recent availability of larger clinical samples of recurring CNVs spurred investigations of individuals who carry the same deletion or duplication irrespective of clinical symptomatology[34].

CNVs at the 22q11.2, 16p11.2, 1q21.1, and 15q11.2 genomic loci are among the most commonly identified risk factors for neuropsychiatric disorders in pediatric clinics[35,36]. These deletions and duplications of genomic sequences strike a balance between occurrence in the population and their strong biological consequences. In other words, while these CNVs are rare, especially compared to SNPs studied in GWAS, these genetic alterations are frequent enough (between 1 in 500 and 1 in 4000 in the general population) so that we can begin to carry out across-CNV investigations in population datasets.

Specifically, these CNVs are now being understood to exert body-wide implications, including in the cardiovascular, endocrine, skeletal and nervous systems[37,38], with deteriorating consequences to everyday life[39]. The substantial downstream consequences suggest that CNVs may serve as a sharp, yet still under-exploited imaging-genetics tool for dissecting the effects of genetic alterations on brain physicality and behavioral differentiation[40]. Although CNVs are well known to impact some of the most lateralized cognitive functions, including language skills[41], how CNVs affect structural brain asymmetry has not yet been explored. 16p11.2 and 22q11.2 deletions stand out as the two "relatively" frequent CNVs with large effects on risk for neuropsychiatric disorders[42,43]. The most substantial increase in risk for schizophrenia (SZ) is associated with 22q11.2 deletions (30 to 40-fold increase)[43], while 16p11.2 deletions confer notably elevated risk (10-fold increase) for autism spectrum disorder (ASD)[42]. In terms of language impairments, 77% of children and 50% of adults carrying a 16p11.2 deletion meet the criteria for childhood apraxia of speech[44]. Furthermore, 95% of children with a 22q11.2 deletion are diagnosed with speech-language disorders[45]. Speech and language delays are hallmark features of CNV carriers in pediatric clinics; perhaps the earliest symptoms for which children with a CNV get clinically referred in the first place[46].

In the present investigation, we have leveraged multiple rare high-effect-size deletions and duplications to interrogate how gene dosage modulates brain asymmetry. Enabled by a multi-cohort machine learning approach, we quantitatively dissected the impact on brain asymmetry across eight CNVs: deletions and duplications at 1q21.1 distal, 15q11.2 BP1-BP2, 16p11.2 proximal, and 22q11.2 proximal loci, with a particular focus on deletions at 16p11.2 and 22q11.2. In 552 CNV carriers and 290 non-carriers, we systematically explored CNV-specific brain asymmetry patterns using a machine learning toolkit, including linear discriminant latent factor modeling. Specifically, we isolated multivariate brain asymmetry patterns that distinguish between respective CNV carriers and controls based on the asymmetry in volumes captured by reference atlases. Given the widely acknowledged associations of CNVs with neuropsychiatric and especially speech-language disorders[35,45], we hypothesized that CNVs affect structural asymmetry in brain regions associated with lateralized functions, including language. Our data-led imaging-genetic results shed new light on reasons why so many CNVs are clinically associated with performance decay in higher cognitive functions, such as language.

## Results
### Comparison of asymmetry patterns across eight CNVs
We aimed to characterize how deletions and duplications of genomic sequences across four genomic loci impact brain asymmetry. For that purpose, we designed an analytic approach yielding eight estimated linear discriminant analysis (LDA) models, each dedicated to a single CNV. In so doing, we isolated multivariate brain asymmetry patterns that discriminate between respective CNV carriers and controls (Table 1). All eight LDA models were effective at performing the CNV-control discrimination task as suggested by ROC analysis (Sup. Fig. 1), where the maximal area under the ROC curve was 0.93 (1q21.1dup), while the minimal was 0.71 (15q11.2dup). Each LDA model included 65 coefficients corresponding to 65 homologous region pairs (48 cortical, 7 subcortical, 10 cerebellar). For each target CNV, the LDA coefficients revealed the contribution of each brain region asymmetry towards the separability of the CNV carriers' brain morphology. We performed bootstrap significance testing (cf. Methods) on the set of eight region-wise LDA effects. Across the total of eight LDA models, CNV-driven left-right asymmetries in the planum temporale and temporal fusiform cortex significantly contributed to distinctive asymmetry patterns in three of the eight CNVs. The temporal fusiform cortex was instrumental in the asymmetry pattern related to 22q11.2 deletion and duplication, as well as 16p2 deletion. The planum temporale asymmetry reached significance for 15q11.2 duplication and the deletions at 16p11.2 and 22q11.2 loci (Fig. 1a).

Overall, the planum temporale reliably showed the strongest effect (i.e., largest absolute LDA coefficient) across all eight CNVs; followed by cerebellum lobule VIIIb and putamen, showing the next

**Table 1 | Dataset demographics**

| Loci | Chr (hg19) start-stop | nGenes (Gene) | Type | Subjects | Age (SD) [range] | Sex (M/ F) | ASD | SZ |
|------|-----------------------|---------------|------|----------|------------------|------------|---------|
| 1q21.1 | chr1 | 7 | Del | 32 | 40 (17) [9–73] | 14 / 18 | 0 | 0 |
| | 146.53–147.39 | CHDIL | Dup | 27 | 44 (15) [8–66] | 11 / 16 | 3 | 0 |
| 15q11.2 | chr15 | 4 | Del | 110 | 55 (7) [40–69] | 50 / 60 | 0 | 0 |
| | 22.81–23.09 | CYFIP1 | Dup | 144 | 54 (7) [40–69] | 67 / 77 | 0 | 0 |
| 16p11.2 | chr16 | 27 | Del | 82 | 19 (15) [7–63] | 48 / 34 | 10 | 0 |
| | 29.65–30.20 | KCTD13 | Dup | 69 | 32 (15) [8–63] | 39 / 30 | 7 | 1 |
| 22q11.2 | chr22 | 49 | Del | 66 | 19 (13) [6–66] | 32 / 34 | 8 | 2 |
| | 19.04–21.47 | AIFM3 | Dup | 22 | 26 (19) [8–66] | 12 / 10 | 2 | 0 |
| Controls | | | | 290 | 26 (14) [6–64] | 162 / 128 | 1 | 0 |

CNV loci chromosome coordinates are provided with the number of genes encompassed in each CNV and a well-known gene for each locus to help recognize the CNV. ASD and SCZ diagnoses for clinically ascertained CNV carriers were obtained from respective data acquisition sites. SCZ diagnosis in the UK Biobank corresponded to the ICD10 code, including schizophrenia, schizotypal, and delusional disorders (F20-F29). ASD diagnosis in the UK Biobank corresponded to the ICD10 code that included diagnoses of childhood autism (F84.0), atypical autism (F84.1), Asperger's syndrome (F84.5), other pervasive developmental disorders (F84.8), and pervasive developmental disorder, unspecified (F84.9).

*Del* deletion, *Dup* duplication, *ASD*: autism spectrum disorder, *SZ* schizophrenia, *chr* chromosome, *Age* mean age, *SD* standard deviation, *nGenes* number of genes.

most pronounced brain effects (Fig. 1b). We, therefore, zoomed in on planum temporale asymmetry in greater detail. Based on post-hoc examination of group difference in planum temporale volume between CNV carriers and controls, deletions at 16p11.2 and 22q11.2, as well as 15q11.2 duplication, significantly increased the asymmetry (two-sided t-test $P_{15q11.2dup} = 0.002$, $P_{16p11.2del} < 10^{-12}$, $P_{22q11.2del} < 10^{-5}$) (Fig. 1c). Conversely, deletions at 1q21.1 significantly decreased planum temporale asymmetry (two-sided t-test P = 0.001). Other CNVs did not display such effects. In sum, CNVs, known to have deleterious consequences on language performance, systematically altered planum temporale asymmetry in our cohort. Therefore, planum temporale, known to be one of the most asymmetrical regions in the human brain[47,48], may also be one of the most susceptible regions to asymmetrical alterations due to genetic mutations.

We observed CNVs to be associated with altered structural asymmetry in different brain regions. To directly probe the similarity between CNV-mediated asymmetry patterns, we quantified the overlap in whole-brain signatures obtained from the eight LDA models using Pearson's correlation across the totality of brain region effects (i.e., model coefficients mapped onto the brain) (Fig. 1d). We used a spin-permutation test across the whole brain followed by FDR correction to examine the statistical significance of Pearson's correlation between each pair of CNV-specific brain maps. The asymmetry brain map pertaining to 16p11.2 deletion showed significant similarity with 1q21.1 deletion (R = 0.35, $P_{spin} = 0.02$), 15q11.2 deletion (R = 0.42, $P_{spin} = 0.01$), 15q11.2 duplication (R = 0.38, $P_{spin} = 0.01$), and 22q11.2 deletion (R = 0.35, $P_{spin} = 0.02$). Similar to prior analyzes of regional volumes, we observed mirroring effects on regional asymmetry conferred by the deletion and duplications at the same loci. The strongest mirroring effect emerged between deletion and duplication at the 16p11.2 locus (R = −0.45, $P_{spin} = 0.01$), followed by the 1q21.1 locus (R = −0.39, $P_{spin} = 0.02$). To summarize these comparisons, we submitted the model coefficients to hierarchical clustering with the Ward similarity measure. This post-hoc analysis revealed three clusters: 1q21.1 and 16p11.2 duplication, our four deletions, 15q11.2 and 22q11.2 duplication. Such separation provides further evidence of the opposing effects of deletions and duplications.

As a final step to chart the commonalities and idiosyncrasies across our panel of eight candidate CNVs, we turned to multiclass LDA modeling as a natural choice to dissect which regions are the most distinctive in the context of the full CNV panel. In contrast to the above analysis, we here estimated a single composite LDA model that considered the full status of all eight CNVs simultaneously with their brain manifestations as a whole. That is, we estimated one LDA model to separate all eight CNVs from each other by solving a multiclass classification problem. Based on the obtained LDA coefficients, the

planum temporale turned out to be the most distinctive region, as documented by the strong contribution to the leading dimension of the multiclass LDA model (Fig. 1e). This finding highlights the effects on planum temporale asymmetries as the most strongly differentiating factor across various genetic alterations. Other regions where the CNV effects differed included the frontal medial cortex, precuneus cortex, frontal pole, or middle frontal gyrus. In summary, planum temporale asymmetry turned out to be a hallmark feature of altered brain asymmetry for some CNVs while not being affected in other CNVs. Therefore, this specific region emerged as the most prominent separating factor among the eight distinct genetic alterations.

## Brain asymmetry effects induced by 16p11.2 and 22q11.2 deletions

After exploring asymmetry patterns associated with a broader portfolio of eight CNVs, we directed our analysis spotlight on two CNVs: deletions at 16p11.2 and 22q11.2 loci as the two most studied CNVs with high penetrance to neuropsychiatric disorders[42,43]. 16p11.2 deletion showed several strong regional asymmetry effects (Fig. 2a). In contrast, 22q11.2 deletion carriers displayed strong LDA coefficients dispersed across the cortex, several subcortical regions, and parts of the posterior lobe of the cerebellum (Fig. 2b). After putting the LDA model coefficients to a bootstrap significance test, 16p11.2 deletion carriers displayed significant negative effects for the planum temporale, lobule VIIIb of the cerebellum, supracalcarine cortex, and putamen. The negative LDA coefficients represent decreased asymmetry in rightward asymmetrical regions (i.e., regions with larger volume in the right hemisphere) or increased asymmetry in leftward asymmetrical regions relative to controls. 16p11.2 deletion carriers further displayed significant positive (larger than zero) coefficients for the parahippocampal gyrus, parietal operculum cortex, cuneal cortex, and temporal fusiform gyrus. The positive LDA coefficients can be interpreted as increased asymmetry in regions with a greater rightward asymmetry or decreased asymmetry in regions with a greater leftward asymmetry. The 22q11.2 deletion was associated with a significant negative coefficient in planum temporale and positive coefficients in the parahippocampal gyrus and temporal fusiform cortex. For both CNVs, the asymmetry patterns derived at the whole-brain level highlighted regions typically thought to subserve lateralized functions, including language, memory, as well as visual, face and word recognition.

## Direction and magnitude of induced asymmetry changes

After identifying the brain regions significantly contributing to the multivariate asymmetry patterns indicative of CNV carriership, we investigated more in-depth the direction and magnitude of the

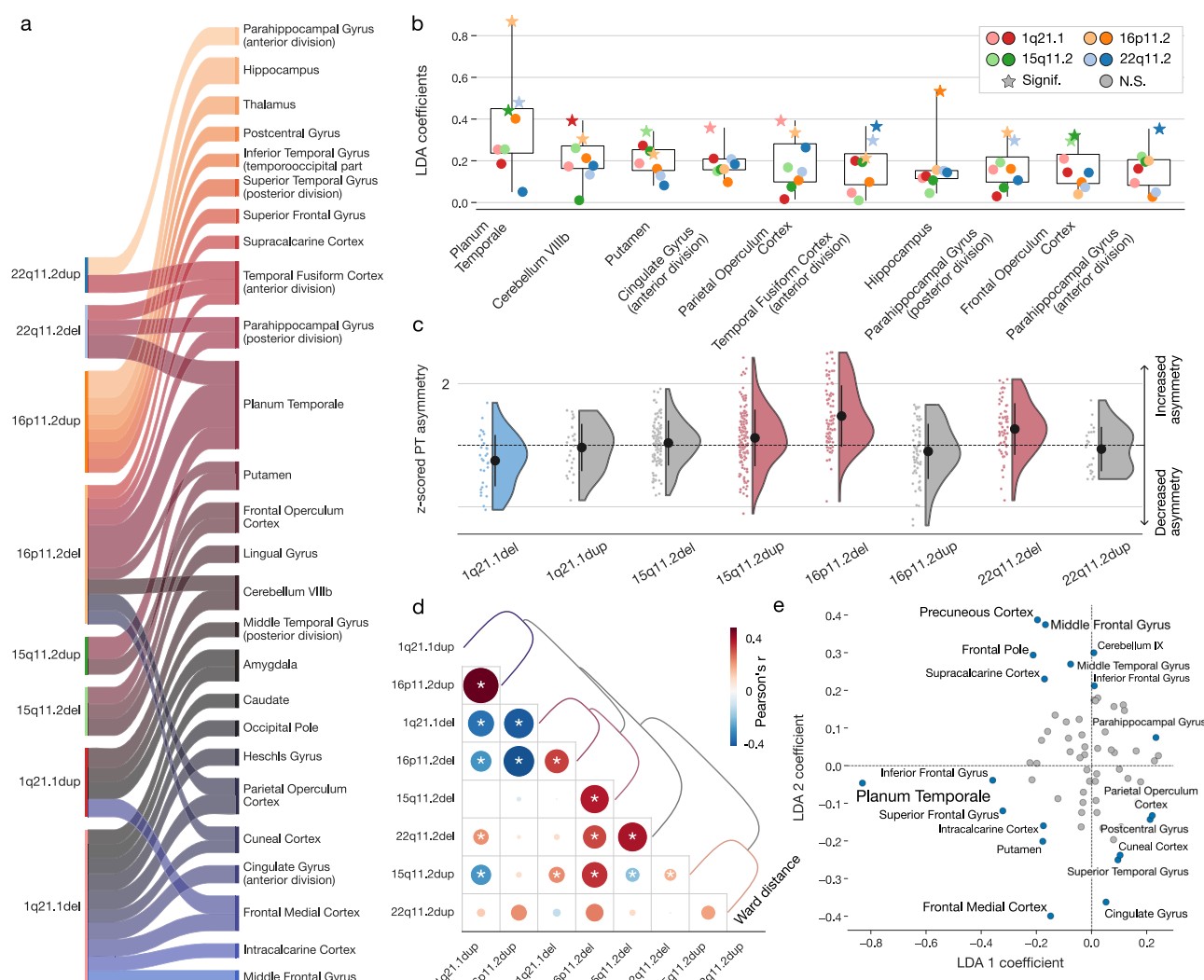

**Fig. 1 | Eight key CNVs lead to unique effects on brain asymmetry that spotlight the planum temporale.** We probed the effects of eight CNVs on brain asymmetry (deletions and duplications at 1q21.1 distal, 15q11.2 BP1-BP2, 16p11.2 proximal, 22q11.2 proximal loci). For that purpose, we estimated eight LDA models encompassing multivariate prediction rules separating respective CNV carriers and healthy controls in terms of regional left-right deviations. **a** Significant LDA coefficients across eight key CNVs. The sankey plot depicts all LDA coefficients surpassing the bootstrap significance test. The width of the ribbon corresponds to the coefficient magnitude. Planum temporale and fusiform cortex asymmetries are both significantly associated with three CNVs. **b** Overall strongest coefficients across eight LDA models. The boxplots depict LDA coefficients across the 8 CNVs (the box extends from the first quartile to the third quartile and the whiskers extend to the 1.5x the inter-quartile range). Lighter symbols represent deletions, while darker symbols represent duplications. Star denotes a significant coefficient based on the bootstrap significance test. **c** Detailing the effects on planum temporale. Rain cloud plots summarize the effects of each CNV on the asymmetry of planum temporale.

The y-axis shows raw asymmetry indices (no LDA model), z-scored using control participants. While the 15q11.2 duplication, 16p11.2 deletion, and 22q11.2 deletion increase the asymmetry, 1q21.1 deletion decreases the asymmetry. The color (blue – decrease, red – increase) depicts a significant change in mean asymmetry based on a two-sided t-test with FDR-corrected $P < 0.05$. **d** Comparison of brain-wide asymmetry patterns. We calculated Pearson's correlation to quantify the similarity between every pair of LDA coefficient sets (vectors) from the eight models. Asterisk denotes FDR-corrected P-values obtained from spin permutation. Hierarchical clustering of model-specific coefficients obtained per CNV display distances (similarities) among studied asymmetry patterns based on Ward's method. Distinct clusters separating deletions and duplications provide further evidence of their opposing effects. **e** Regions separating different CNVs. Region-wise coefficients are plotted along the two leading components of the multiclass LDA model designed to separate between the eight CNVs. CNVs lead to distinct brain asymmetry patterns in which the planum temporale plays a prominent role.

discovered differences in volume asymmetry. 16p11.2 and 22q11.2 deletion carriers displayed significant negative LDA coefficients for the planum temporale. Since the planum temporale, an insular region behind Heschl's gyrus, is typically larger in the left hemisphere in neurotypical individuals[47,48], the negative coefficient indicated widened planum temporale asymmetry in CNV carriers (Fig. 3a). In other words, based on our local asymmetry region measures (i.e., asymmetry index), carrying a 16p11.2 or 22q11.2 deletion resulted in an increased leftward asymmetry of the planum temporale compared to controls. Both CNV carriers also displayed decreased asymmetry in the

temporal fusiform cortex and parahippocampal gyrus. That is, the asymmetry of these two regions was significantly reduced in subjects with deletions at 16p11.2 and 22q11.2 loci. In addition to the above-described effects from both CNVs, 16p11.2 deletion carriers also displayed a significant increase in parietal operculum and cuneal cortex asymmetry. Finally, carrying a 16p11.2 deletion resulted in a smaller divergence between left and right hemispheric volumes in the putamen, supracalcarine cortex, and lobule VIIIb of the cerebellum.

Deviations from normal brain asymmetry induced by the genetic alterations probably arise from separate impacts on each of the two

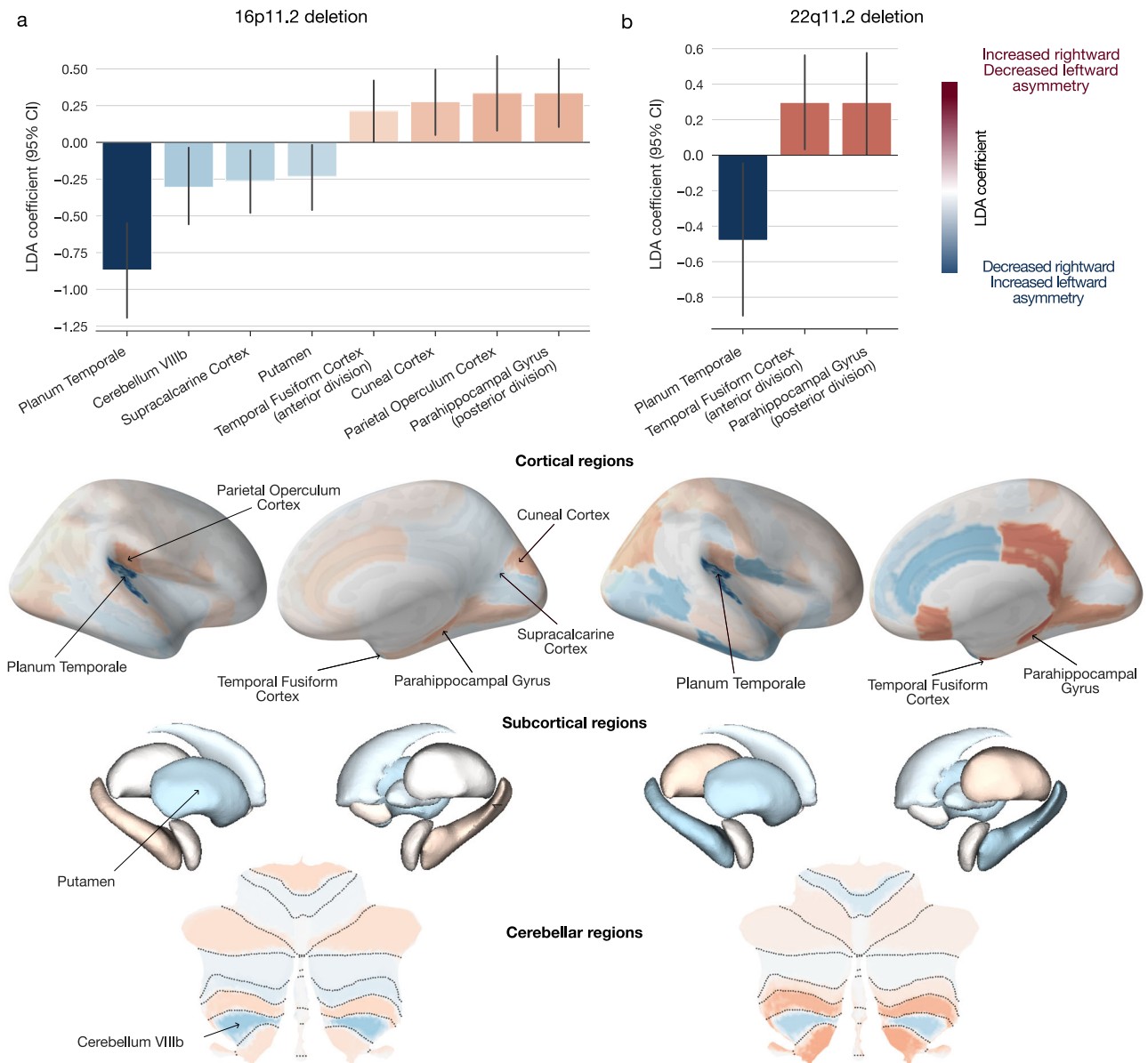

**Fig. 2 | Genomic deletions at 16p11.2 and 22q11.2 loci lead to distinctive brain asymmetry patterns.** For deletions at the 16p11.2 and 22q11.2 loci, we estimated two dedicated LDA models to separate between the carriers of a particular CNV and controls based on 65 left-right asymmetry measures of brain atlas volumes. Each ensuing LDA model encompassed 65 coefficient estimates that pertained to 65 brain regions that quantify the contributions to the separating multivariate asymmetry patterns. **a** LDA-derived brain asymmetry effects for 16p11.2 deletion. The bar plot depicts the eight high-effect-size regions that display significant LDA coefficients based on bootstrap testing. The negative planum temporale coefficient stands out as the strongest coefficient across cortical, subcortical, and cerebellar structures. **b** LDA-derived brain asymmetry effects for 22q11.2 deletion. This mutation is associated with several strong LDA coefficients across the brain, with three coefficients passing the bootstrap testing. Positive (red) LDA coefficients represent decreased leftward asymmetry or increased rightward asymmetry, and vice versa for negative (blue) coefficients. Error bars correspond to the 95% confidence interval (CI) based on bootstrap resampling distribution with 1000 iterations. Deletions at 16p11.2 and 22q11.2 loci lead to asymmetry deviation principles involving regions typically associated with higher-order cognitive functions.

homologous regions. Hence, for each brain region in our reference atlas, we next conducted a separate case-control comparison of CNV effects on local regional volume: one region-wise group contrast in the left and another one in the right hemisphere. We thus computed Cohen's $d$ between CNV carriers and controls separately for the left and right hemispheric volume measures of a particular atlas region (Fig. 3b) that we previously identified as discriminatory using LDA (Fig. 2). In addition, we assessed the robustness of each Cohen's $d$ measure using bootstrap-derived 95% confidence interval (cf. Methods). Based on this descriptive post-hoc examination of the brain region volumes, we observed the largest effect sizes for planum temporale in the left hemisphere for both 16p11.2 and 22q11.2 deletion

carriers. Specifically, planum temporale showed a larger raw volume in 16p11.2 deletion carriers compared to controls in both hemispheres. This case-control difference was more prominent in the left hemisphere compared to the right hemisphere (Cohen's $d_{left} = 1.39$, 95% CI [1.06, 1.72]; Cohen's $d_{right} = 0.60$, 95% CI [0.30, 0.87]). A larger case-control difference for the planum temporale in the left hemisphere was also present in 22q11.2 deletions (Cohen's $d_{left} = 0.76$, 95% CI [0.50, 1.02]; Cohen's $d_{right} = 0.24$, 95% CI [0.02, 0.50]). As another example, 16p11.2 deletion carriers showed lower volumes of temporal fusiform cortex than controls. The effect size was significant only in the left hemisphere and not in the right hemisphere (Cohen's $d_{left} = -0.59$, 95% CI [−0.87, −0.32]; Cohen's $d_{right} = -0.16$, 95% CI [−0.41, 0.10]). We

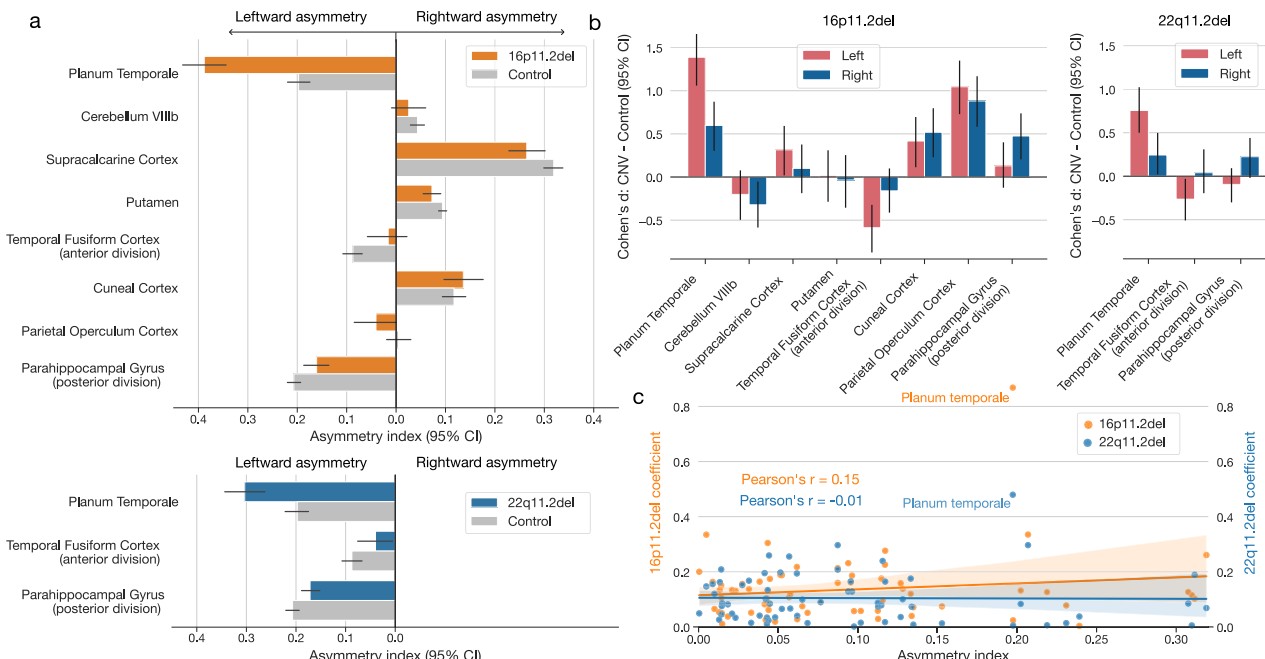

**Fig. 3 | Zooming in on the left- and right-biased effects to explain asymmetry disbalances in two CNVs.** We detailed the effects of CNV-carriership on asymmetry by examining the raw hemispheric volumes in regions identified as significant (Fig. 2). **a** Asymmetry index in CNV carriers and controls for deletions at 16p11.2 and 22q11.2 loci. For both loci, CNV carriers show the highest increase in planum temporale asymmetry compared to controls. Both CNVs also show decreased asymmetry for the fusiform cortex and parahippocampal gyrus (error bars denote a 95% confidence interval for the mean asymmetry index). **b** Disambiguating the direction of left-biased vs right-biased effects. The bar plot depicts differences in regional volumes between CNV carriers and controls in each hemisphere. 16p11.2 deletion displays a similar direction of left- and right-biased effects. In other words, this CNV

impacts both hemispheres in the same direction for each regional volume. However, the magnitudes of volumetric changes differ between each hemisphere. 22q11.2 deletion leads to significant effects in a single hemisphere. Specifically, this CNV decreases volume in the left fusiform cortex and increases the volume in the right parahippocampal gyrus. **c** Association between regional asymmetry and CNV effects. The scatterplot depicts the relationship between the coefficients of CNV-specific models and regional asymmetry in controls. Both 16p11.2 and 22q11.2 deletions effects do not significantly correlate with regional asymmetry (Pearson's correlation $P > 0.05$). Planum temporale stands out as the most affected region in both CNVs. Volumetric effects leading to asymmetry alteration depend on brain region and type of genetic mutation.

observed the same impact on this region for the 22q11.2 deletion (Cohen's $d_{left} = -0.26$, 95% CI [−0.50, −0.03]; Cohen's $d_{right} = 0.04$, 95% CI [−0.20, 0.30]). These results highlight how the magnitude of volumetric alterations differed between carriers and non-carriers for each brain hemisphere.

As a final step in this detailed exploration of asymmetry changes, we examined whether these CNV-specific changes are associated with regional asymmetries measured in controls. We computed Pearson's correlation between the two sets of LDA coefficients and regional asymmetry averaged across all controls (Fig. 3c). For both CNVs, the relationship was not significant ($P_{16p11.2} = 0.24$, $P_{22q11.2} = 0.93$). The non-significant association demonstrated that CNV effects are highly specific and independent of the original regional asymmetry. Taken together, 16q11.2 and 22q11.2 deletions impact brain asymmetry in similar parts of the brain. These alterations arise from different magnitudes of left versus right bias in the central nervous system.

### Functional profiling of CNV asymmetry brain signatures

We subsequently turned to NeuroSynth[49] to mine candidate functional explanations for the distilled brain asymmetry patterns that separated CNV carriers from controls. To this end, we mapped the full collection of 65 (absolute) LDA coefficients onto both hemispheres, with the same coefficient for homologous regions. In other words, we created a whole-brain map indicative of CNV carriership based on anatomically localized classifier coefficients. Then, these LDA-derived brain representations were contextualized based on a curated collection of thousands of functional activation maps hosted by the NeuroSynth resource. This automated annotation framework yielded keywords sorted according to the strength of association with the brain

representation pertaining to each CNV. We identified cognitive and psychological domains (among 3,228 candidate terms) most strongly associated with each CNV status.

The whole-brain constellation of left-right asymmetry shifts driven by 16p11.2 deletion was robustly associated with terms such as *primary auditory*, *Heschl's gyrus*, *planum temporale*, *sound*, *tones*, and *acoustic* (Fig. 4a). The terms related to brain anatomy recapitulated the topographical hotspots of brain asymmetry differences (cf. above). We further refined the list to only contain concepts related to the brain and cognition. In this sub-analysis, functional decoding related 16p11.2 deletion to the notions of *sounds*, *acoustic*, *noise*, *tones*, *audiovisual*, *speech perception*, or *listening*. The 22q11.2 asymmetry brain pattern was associated with a collection of terms such as *Heschl's gyrus*, *parahippocampus*, *primary auditory*, or *auditory cortex* (Fig. 4b). Similar to 16p11.2 deletions, certain flagged keywords readily reflected the spatial distribution of LDA-derived CNV effects across the brain. After discarding the terms related to brain anatomy, the strongest brain-ontology associations featured the terms *dementia*, *tones*, *acoustic*, *sounds*, *psychophysical*, *noise*, and *posttraumatic*. Quantitatively, the morphology-term associations for the brain asymmetry patterns of 16p11.2 and 22q11.2 deletions were similar, as evidenced by the strong Pearson's correlation between the NeuroSynth association strengths (R = 0.99) (Fig. 4c). Taken together, across the two examined CNVs, we observed that LDA-derived asymmetry brain representations were mainly tied to capacities revolving around language and hearing, both of which are clinical hallmark features of these genetic mutations in pediatric healthcare institutions[50,51].

Based on an even more specific pool of functional NeuroSynth descriptions, we explored the distribution of cognitive functions along

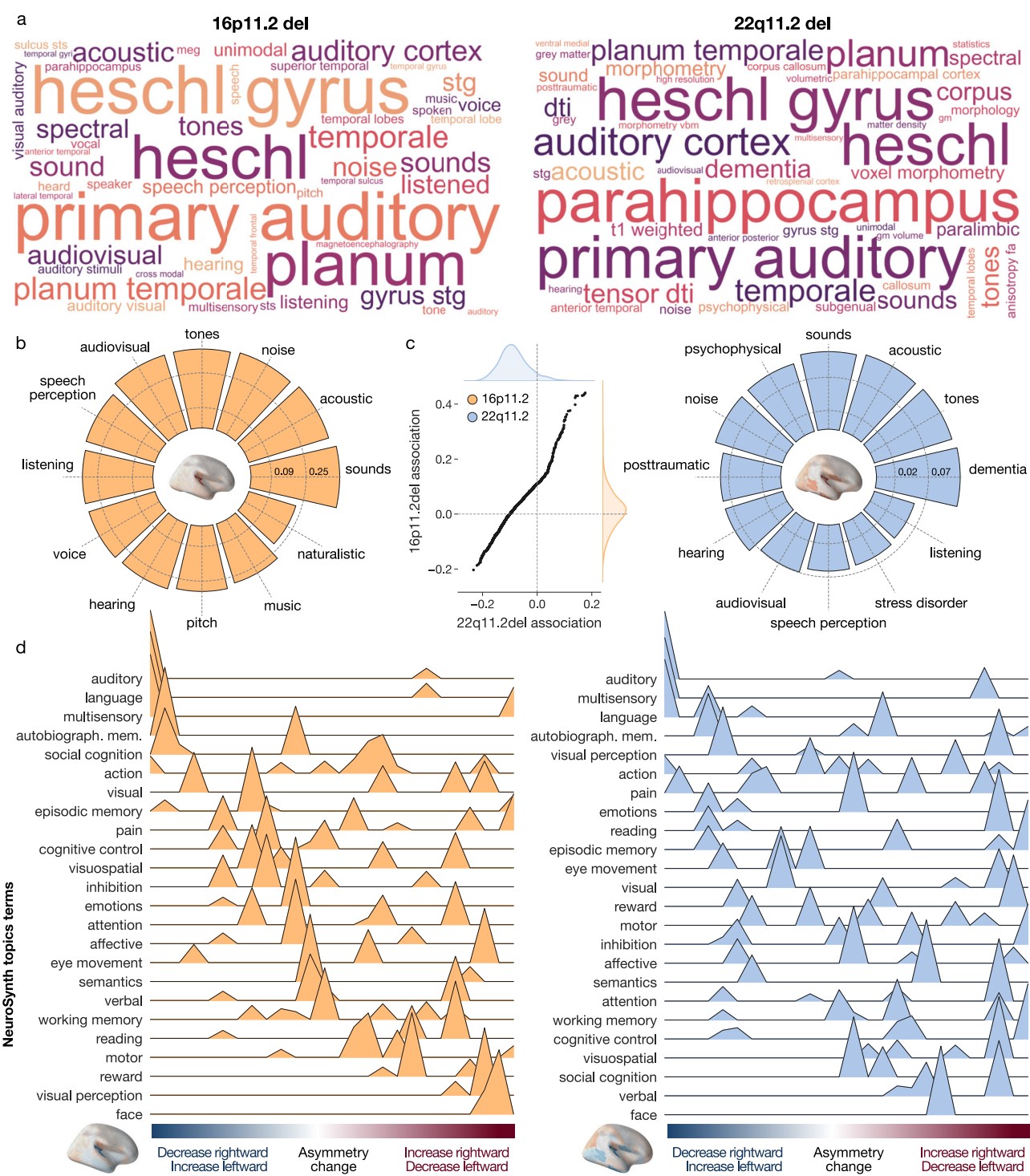

**Fig. 4 | Functional decoding of derived asymmetry patterns spotlights language and hearing.** We performed a functional profiling assay based on the obtained LDA asymmetry patterns using the NeuroSynth resource. To that end, we first map the 16p11.2 and 22q11.2 deletion LDA coefficients on the brain. Resulting whole-brain signatures summarize the asymmetry differences between controls and respective CNV carriers. We then contextualized the whole-brain asymmetry signatures by means of curated NeuroSynth activation maps. **a** Top associated NeuroSynth keywords. Word cloud plot depicts the 50 most strongly associated NeuroSynth keywords for each of the two CNVs. **b** Functional associations with LDA patterns. Circular bar plots display the 12 most strongly associated keywords after filtering out terms related to brain anatomy. Y-axis corresponds to the similarity (Pearson's correlation) between the brain map of LDA coefficients and the respective NeuroSynth activation map. **c** Similarity in functional profiles of both

CNVs. Each axis shows the correlation between the 3228 term-specific NeuroSynth activation maps and a given CNV-specific asymmetry pattern. The strong linear relationship demonstrates that both CNVs are associated with similar functions. **d** NeuroSynth meta-analysis along ranked LDA coefficients using 24 topic terms. We calculated the weighted score of activation maps and ranked LDA asymmetry patterns. Terms are ordered by the weighted mean of their location along the LDA coefficient spectrum. Negative LDA coefficients (blue) correspond to decreased leftward or increased rightward asymmetry, and vice versa for positive coefficients (red). The negative coefficients are associated with language and hearing for both CNVs. The positive coefficients are associated with face processing and visual perception. 16p11.2 and 22q11.2 deletion impact the laterality of higher-order functional systems, including language, hearing, and visual perception.

the spectrum of LDA coefficients. Specifically, we examined the association between a list of topic terms with CNV-induced asymmetry changes binned into five-percentile increments (cf. Methods). For the 16p11.2 deletion, negative LDA coefficients – which represent decreased rightward asymmetry or increased leftward asymmetry – were associated with *language*, *auditory*, and *multisensory* functions (Fig. 4d). Positive LDA coefficients, which reflected reduced leftward asymmetry or increased rightward asymmetry, were associated with *visual processing* and *face perception*. Analysis of the 22q11.2 deletion yielded a similar constellation of results. As such, we showed that regions with strong negative coefficients are associated with the perception and production of *language* and *auditory* cues. In contrast, regions with strong positive effects were associated with *vision functioning* and *social cognition*. As such, our collective findings highlight how the increase in the leftward asymmetry of the planum temporale is associated with language production. On the other hand, the decrease in the leftward asymmetry of the temporal fusiform cortex was associated with visual perception. Our results thus provided a link between the direction of CNV-induced changes in brain asymmetry and their impact on designated neural systems.

To supplement the functional profiling using NeuroSynth, we investigated if derived asymmetry patterns are associated with handedness. Although the contributions of genetic variants to handedness remains a conundrum[52], recent studies point to the role of rare variants[53]. Therefore, we probed for the association between the derived asymmetry pattern and handedness in the 36,000 UK Biobank participants. Nevertheless, we did not find any robust association (Sup. Fig. 2).

## Downstream consequences of common variant genetics on brain asymmetry

Our analysis of rare high-effect CNVs spotlighted the asymmetry of planum temporale structure to be especially vulnerable to such genetic alterations. To complement this local genetic analysis based on CNVs with the influence of common variants (SNPs), we conducted a GWAS (genome-wide association study) of planum temporale volume asymmetry in the 29,470 UK Biobank imaging participants.

Planum temporale asymmetry turned out to relate to a single locus on chromosome 10p14 (rs41298373, $P = 3.31 \times 10^{-17}$) (Fig. 5a). This single significant SNP is a coding non-synonymous variant with a particular LD structure reflected by the lack of other SNPs in this locus. The SNP significance was confirmed by several sensitivity analyzes directed at the definition of the asymmetry index (Sup. Fig. 3). The associated quantile-quantile (QQ) diagnostic suggested that future GWAS based on larger sample size is not likely to reveal other SNPs (Fig. 5b). The identified rs41298373 at this locus represents a coding polymorphism within exon 9 of gene *ITIH5* (ENSG00000123243) (Fig. 5c). *ITIH5* (inter-alpha-trypsin inhibitor heavy chain 5) is a protein-coding gene previously reported to be involved in extracellular matrix stabilization and the prevention of tumor metastasis[54]. Genetic studies listed in the GWAS Catalog[55] associated the rs41298373 locus with various measures of brain structure, such as sulcal depth, left-right brain asymmetry, cortical surface area, and cortical thickness[28,56–58] (Fig. 5d). In addition to previous GWAS associations with the rs41298373 SNP, the gene *ITIH5* was further linked to blood protein or gut microbiome measurements that included serum levels of proteins ITIH2, ITIH1, NPPB, SIRT2 as well as Ruminococcaceae abundance or ovarian reserve[59–61] (Fig. 5e). Collectively, these observations suggest potential downstream consequences on multiple measures of brain morphology for this gene variant.

Next, we explored if there are similar genome-wide association profiles (including the significant association with the *ITIH5* gene) for the raw volumes of the left and right planum temporale (Sup. Fig. 4). Notably, the planum temporale asymmetry-related *ITIH5* gene was only observed in SNPs associated with the left planum volume

(rs41298373 $P = 5.64 \times 10^{-13}$), but not the right planum temporale. Based on GWAS of the left planum temporale volume, we found 726 significant candidate SNPs that mapped to eight genomic loci. After functional filtering for gene mapping, we obtained eight lead SNPs that are assumed to entail variations of ten protein-coding genes (*CDC42, FAM172A, ITIH5, BCL11B, LRCH1, ENPP2, SEMA3D, WNT4, TLN2, KIAA0825, WNT4, CDC42, FAM172,* and *KIAA0825*) (Fig. 5f).

Incongruent with the GWAS signals pertaining to the left planum temporale, our GWAS of the right planum temporale volume yielded 368 candidate SNPs in four genomic loci. We further refined this SNP set to four lead SNPs related to six genes (*DMRTA2, FAF1, ATP1B3, TFDP2, C14orf177,* and *C14orf64*) – none of which was related to left planum temporale volume (Fig. 5f). Notably, we observed more SNPs associated with the left planum temporale compared to its right hemispheric homolog. Such an observation is consistent with larger CNV effects on left planum temporale.

Since we observed that all three brain structural phenotypes under study – the left and right volumes, as well as the asymmetry of planum temporale – are associated with different SNPs, we aimed to quantify the degree of overlap in their genetic bases (i.e., their associated summary statistics). We found a notable yet incomplete genetic correlation between the genetic underpinnings that influence the left and right planum temporale volumes in our UK Biobank participant sample ($R_g = 0.85$, $P = 1.53 \times 10^{-128}$) (Fig. 5g) – an identical value as prior research[56]. However, this result also demonstrated that there was still unexplained variance, which suggests that certain GWAS signals are not shared between the two phenotypes. In other words, genetics bases for left and right planum temporale volumes in adults were partly diverging. Regarding the genetic architecture, the volume of left planum temporale showed a stronger genetic correlation with planum temporale asymmetry ($R_g = 0.50$, $P = 4.80 \times 10^{-10}$) compared to the volume of right planum temporale ($R_g = -0.02$, $P = 0.87$). Therefore, the genetic architecture of planum temporale asymmetry bore a degree of similarity to the GWAS signals pertaining to the left planum temporale volume but not those of the right planum temporale volume.

To complement these findings, we also quantified the genetic correlation with ASD and SCZ (cf. Methods). Genetic correlations with ASD (asymmetry: $R_g = 0.12$, left volume: $R_g = 0.11$, right volume: $R_g = 0.05$) were generally stronger than with SCZ (asymmetry: $R_g = 0.003$, left volume: $R_g = -0.03$, right volume: $R_g = -0.03$). However, none of the correlations reached significance (minimal p-value = 0.31 for the genetic correlation between ASD and planum temporale asymmetry). In conclusion, across several different genetic analyzes, we found evidence of partly diverging genetic influences on the structure of the right versus left planum temporale.

Our collective analyzes demonstrate that rare CNVs impact brain asymmetry of specific brain regions, and especially those involved in higher-order cognition brain systems. To support this conclusion, we isolated brain asymmetry patterns from the collection of eight key CNVs. Furthermore, we provided a detailed description of the regional effects driving abnormal brain asymmetry. Finally, we carefully annotated the distilled asymmetry patterns by means of large-scale functional profiling. Taken together, our findings highlighted brain asymmetry as a crucial intermediate phenotype on the causal pathway from recurrent genetic alterations to advanced human faculties.

## Discussion

Rare CNVs are an emerging device in the toolkit for studying how specific genetic alterations reverberate in the human brain and phenome. In this pattern-learning study, we carefully quantified the ramifications of eight key CNVs with the goal of revisiting a major design principle of human brain organization – asymmetry between the left and right hemispheres. We draw a detailed picture of anatomical changes, functional associations and relevant genetic

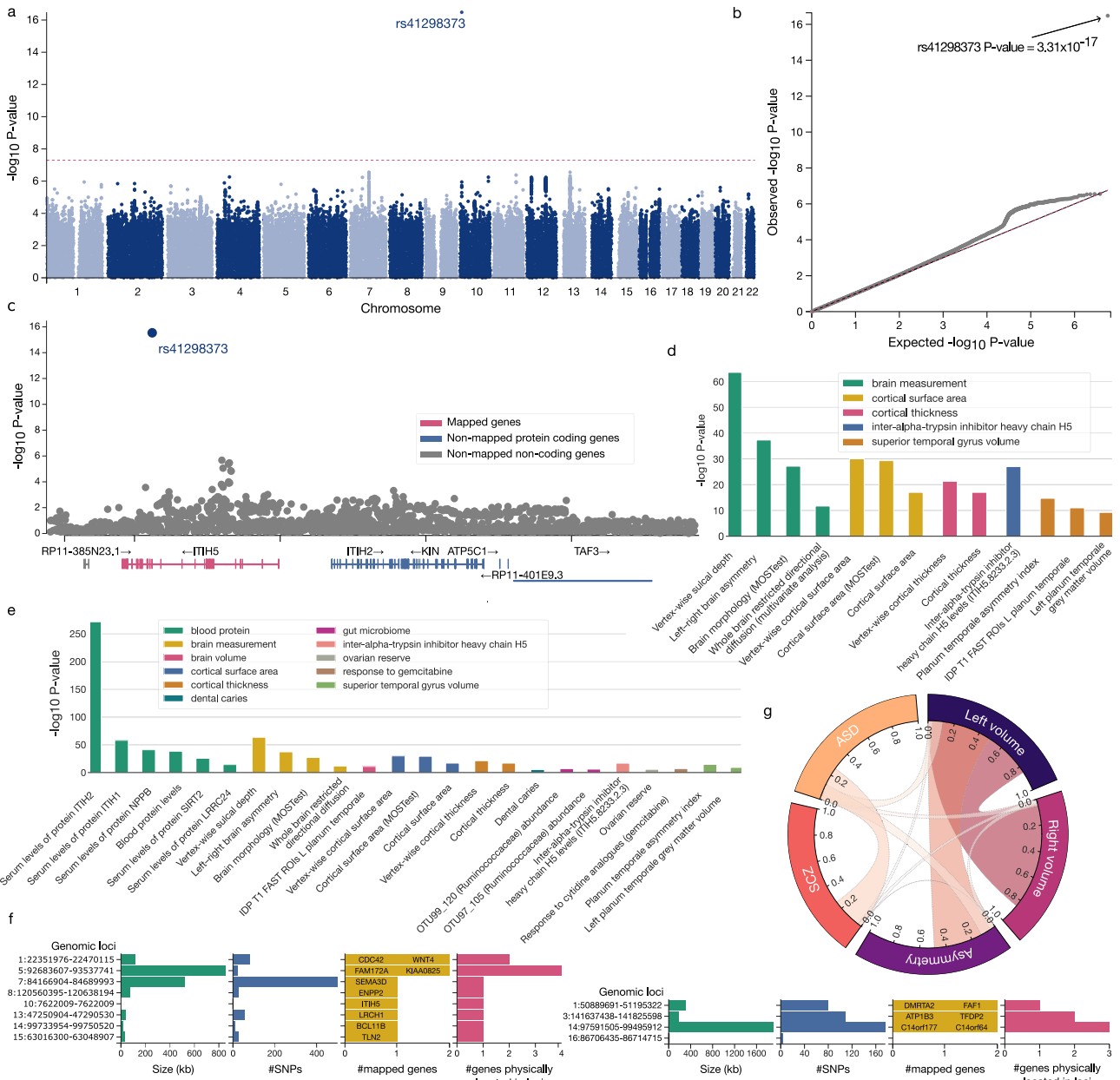

**Fig. 5 | Single common variant significantly associated with planum temporale asymmetry in genome-wide scanning.** We performed a genome-wide association study (GWAS) based on 29,470 UK Biobank subjects to find associations between common single nucleotide polymorphisms (SNP) and planum temporale asymmetry. **a** GWAS of planum temporale asymmetry spotlights single SNP. The Manhattan plot depicts the single significantly associated SNP rs41298373 at locus 10p14. **b** Quantile-Quantile plot for the performed GWAS. The *P*-value associated with rs41298373 clearly exceeds the expected *P*-values from the corresponding reference distribution. **c** Regional plot to zoom in on the identified genomic locus rs41298373. The identified SNP locus lies within exon 9 of the gene ITIH5 (ENSG00000123243). **d** GWAS associations with rs41298373 based on the GWAS Catalog. This SNP was further associated with several measurements of brain morphology or cortical surface and thickness. In total, previous GWAS associated rs41298373 with 13 phenotypes, including the here observed planum temporale asymmetry (i.e., second bar). **e** GWAS associations with *ITIH5* based on the GWAS Catalog. The identified gene *ITIH5* was further associated with a total of 24 phenotypes spanning measurements of blood proteins, brain morphology, gut

microbiome, or ovarian reserve. **f** Summary of GWAS based on left, respectively right, planum temporale volume. GWAS of left planum temporale identified significantly associated SNPs influencing the expression of 10 genes, including the *ITIH5* gene. GWAS of right planum temporale volume identified SNPs associated with the expression of six genes. Notably, *ITIH5* was not absent among these genes. **g** Genetic correlation between genetic basis (GWAS summary statistics) underlying left and right planum temporale volume, planum temporale asymmetry, as well as ASD and SCZ. The left and right planum temporale volumes were significantly, yet imperfectly correlated. The remaining unexplained residual variance suggests a partly diverging genetic architecture. The genetic architecture underlying planum temporale asymmetry was significantly correlated only with the volume in the left hemisphere but not that of the right one. We did not find a significant correlation with any of the disorders. Plots were generated using FUMA software[158]. We isolated a specific genetic locus that may mediate genetically controlled brain asymmetry. In addition, we quantified the different genetic control of left and right planum temporale volume.

underpinnings of region alterations that underlie the left-right imbalance induced by CNV carriership. Based on thorough profiling and comparison of CNV effects on brain morphology, we identified the planum temporale as a key region that is susceptible to this class of circumscribed, recurring genetic alterations. This converging result motivated a post-hoc GWAS investigation of the planum temporale volume, which confirmed that the genetic basis of left and right planum temporale volume is partly diverging.

Among the distilled morphological patterns, the planum temporale asymmetry played a central role. More specifically, the planum temporale was the most commonly and most strongly affected region in our assessed palette of eight CNVs. Prior studies identified volumes of left and right planum temporale as sensitive to gene dosage effects and as a source of morphological variation across CNVs[62,63]. Planum temporale is a triangular-shaped region which occupies the superior temporal plane posterior to Heschl's gyrus[10]. It is part of a neural circuit that passes through left-hemisphere language regions[3]. This region forms a part of the perisylvian area that is known to show the greatest structural left-right differences in the human brain in general[6]. Hence, in neurotypical individuals, the planum temporale is highly structurally lateralized, with 65% of all individuals showing larger left planum temporale; and only 11% of individuals showing larger right planum temporale volume[47,64]. In terms of functional lateralization, there is a possibility that microstructural and not macrostructural features are more important[65]. Postmortem studies highlighted the role of intrinsic microcircuitry organization[66]. However, the role of micro- and macrostructure is beyond the scope of this article since the structural MRI signal captures contributions from various structural components within a voxel and is not able to disentangle microstructure properties[67]. Of note, we found that CNVs can lead to both increased and decreased planum temporale asymmetry, suggesting that different kinds of deviations might be associated with language performance. However, it is important to note that planum temporale asymmetry is not a marker of inter-individual variability in language dominance[68].

Clinical evidence indicates that lesions involving the left superior temporal gyrus, which extends into the planum temporale, give rise to Wernicke's aphasia. This tissue impairment typically leads to fluent but disordered speech production, impaired understanding of others' speech, and impaired silent reading[69,70]. Further brain lesion and brain-imaging studies confirmed that the planum temporale is implicated in several crucial aspects of human communication, both spoken and gestural forms, that rely on a number of component neural processes, such as auditory and phonological processing, language comprehension[71] and subvocal articulation[72,73]. A larger hemispheric divergence of the planum temporale surface was found among dyslexic children[10]. In addition, planum temporale surface asymmetry was also related to a family history of dyslexia[74]. Finally, reduced activity related to processing speech sounds in the planum temporale has been observed in children with developmental dyslexia[75]. Despite the necessity of the planum temporale for realizing language- and speech-related capacities, the planum temporale is currently viewed as a general advanced auditory signal processing hub responsible for representing and binding the location of sounds in space[76] as well as for template-matching operations between configurations of incoming sounds and previously stored auditory signals[72]. Therefore, planum temporale circuits probably help solve a general neural computation that operates over many classes of stimulus types, including simple sound patterns[77], spectro-temporal modulations[78], speech, speech-like sounds, and speech-related cues such as prosody[79-81], but, notably, also tonal perception in music[82-84]. Indeed, a prior MRI study found increased cortical thickness and gray matter density of planum temporale in musicians[85]. Consistently, our functional decoding analysis associated patterns centered on planum temporale asymmetry with functional keywords such as *music, pitch*, and *tones*. Therefore, our results suggest that CNVs impacting planum temporale asymmetry have downstream consequences on several lateralized functions highly evolved in humans, such as language and advanced auditory perception like music or voice prosody appraisal.

In addition to having the probably most drastic functional asymmetry between the left and right brain, we are the only species that has given rise to a culture fueled by communication via articulated language[86]. In our communication-centered society, language facilitates cultural inventions and has enabled many of today's societal institutions. An integral mode of exchange between humans is the "universal language" of music. It is a cornerstone on which our feeling of togetherness and community rests. That is why, at the center of many, if not all, cultures studied so far is music a fundamental part of social customs and rites – a cultural cornerstone[87]. Music is thought to be the "social glue" that enhances cooperation by strengthening feelings of unity and social belonging[88,89]. Besides its cohesive benefit for our society, music can facilitate and intensify bonds in large groups and communities – a possible evolutionary extension of grooming in monkey societies[90]. These elements illustrate why language and music have probably played important roles in the adaptation that the human brain and its genetic control underwent in evolutionary times. Notably, prior studies have shown that CNVs may affect music perception[91]. In summary, it may be no accident that some of the most asymmetrical parts of the brain are also related to some of the most human-defining cognitive capacities in general. In this scenario, higher-order auditory functions and music-related practices are two prominent examples.

The role of language in developmental periods is fundamental, from prenatal development[92] to adulthood. Thus, language impairments will have lifelong consequences for the ability to meet the demands and challenges of contemporary societies. As an example, language impairment may contribute to reported lower educational attainment and decreased ability to earn income in adult life in CNV carriers[39]. Furthermore, language-related symptoms are present in many neurodevelopmental disorders. CNVs at our four loci present some of the most frequent risk factors for these disorders (up to a 40-fold increase in the risk[43]). A line of evidence shows that the sources of these specificities co-localize to the planum temporale. A reduction of planum temporale gray matter volume asymmetry has been reported in people with ASD[93,94]. A reduction of the left planum temporale has been proposed to relate to delays in language acquisition, which are commonly observed clinical features in ASD patients[95]. Consistent with these findings, a study in newborns reported a strong association between the growth rate in the left planum temporale over the first 3 weeks of life and language scores at 12 months of age[96].

On the clinical ward, language impairments represent a hallmark feature of CNV carriers in the majority of medical genetics clinics for children. CNV-induced language impairments include the absence of expressive language, problems with receptive language, compromised prosodic functions, and ASD-like communication deficits[97-99]. Moreover, the impairments co-occur with speech disorders such as childhood apraxia of speech, dysarthria (speech muscle weakness), dyslexia, or nonverbal oral apraxia[100,101]. Prior CNV research could not yet fully explain why language is typically impaired in these patients. We here offer a neuroscientific basis for this frequent clinical observation, with its detailed characterizations in the brain and phenome. Moreover, we highlighted the consequences of the identified impairments to social life, as reflected in the difficulties of forming social bonds due to communication problems[102].

The average leftward asymmetry of the planum temporale is established even before birth. It can be observed starting from the 31st week of gestation, as evidenced by *post-mortem* studies of fetuses[103]. Furthermore, perisylvian regions, including the planum temporale, were the only regions to be asymmetric based on in-vivo MRI of pre-term newborns from 26 to 36 weeks of gestational age[104]. Given this repeated observation, average leftward asymmetry of the planum

temporal appears to be encoded in the human genome, yet by currently unknown mechanisms. Although detailed knowledge of brain structure development in CNV carriers over the lifespan is lacking, macrocephaly was observed already in utero in a 1q21.1 distal duplication carrier[105]. A recent study pointed to a heritability of planum temporale structural asymmetry of around 14%[56]. Prior GWAS research of low-effect genetic variants found two loci (rs7420166, rs41298373) in 18,000 subjects. These two loci affect the expression of the genes *ITIH5* as well as *BOK* and *DTYMK*, respectively. Similar to our results, the authors did not find significant genetic correlations of planum temporale asymmetry with autism spectrum disorder, attention deficit hyperactivity disorder, schizophrenia, educational attainment, or intelligence using the GWAS summary statistics. This is despite evidence for genetic overlap between brain asymmetries and autism, educational attainment, and psychiatric disorders with language impairment[28].

To complement our analysis of local CNVs on brain morphology, we also probed the genome-wide effects underlying planum temporale variation on a much larger dataset. Specifically, we extended the study to almost 30,000 UK Biobank participants. Our GWAS of planum temporale asymmetry confirmed a specific SNP (rs41298373), which was reported previously[56]. Notably, our whole-genome results confirm previous findings that this locus is significantly associated only with the left, but not the right, planum temporale volume. GWAS on the left planum temporale volume revealed a set of ten significantly associated loci (genome-wide significant lead SNPs). These SNPs differed from the six SNPs identified through GWAS on right planum temporale volume. The different sets of SNP hits and an imperfect genetic correlation of 0.85 confirm that the genetic basis underlying left and right planum temporale volumes is partly diverging. Since the planum temporale is asymmetrical already in the mother's womb, we here isolate genetic loci whose downstream protein products may mediate this genetically controlled brain asymmetry before the infant has interacted with the external world.

Along with planum temporale, our whole-brain analysis also spotlighted CNV-mediated brain asymmetry in the parahippocampal gyrus, the temporal portion of the fusiform gyrus, or cerebellum lobule VIIIb, whose volumes were susceptible to the examined genetic alterations. Most of the identified brain regions are associated with higher-order functions such as memory encoding and retrieval or visual processing and recognition. As evidenced by the strong effects on cerebellum lobule VIIIb, the CNV effects are not limited to the cerebrum. A prior study found several cerebellar regions (vermis lobule VIII-X and cerebellar cortex) to be highly sensitive to CNVs – potentially due to the cerebellum's protracted development[106]. Notably, individuals with the most functionally asymmetric cerebrums also appear to possess the most asymmetric cerebellums in general[107]. Within the cerebellum, the most strongly leftward asymmetrical regions are located in lobules VI and VIII[107]. The lobule VIII is mostly part of the advanced processing cerebellum that contributes to higher-level cognition[108]. Prior studies reported the involvement of this region in somatosensation[109] and motor tasks[110]. However, the volume of lobule VIIIb was also associated with advanced cognitive capacities, including reading as well as mental rotation and inhibitory control[111,112]. In addition, this subcompartment is adjacent to the lobule VIIIa, in which increased gray matter volume was linked to better scores on language measures in several individual studies[113,114] and in meta-analytical evidence[115]. Therefore, we provide another brick of evidence that the currently under-appreciated cerebellum – which encompasses distinctive structural and functional compartments – could participate in functions uniquely developed in humans.

Another region that played a key role in our distilled asymmetry patterns was the parahippocampal gyrus. This part of the temporal lobe plays an important role in spatial memory[116] and navigation[117] processes. These two functions are concrete examples of this structure's general abstract space mapping capacity. Moreover, the mapping capacity may extend to semantic representational spaces with possible relevance to language and social representational spaces[118,119]. Finally, our whole-brain asymmetry analysis also spotlighted the temporal fusiform cortex to be associated with 16p11.2 and 22q11.2 deletions. Fusiform gyrus was reported to be among regions markedly altered across CNVs[63]. From a functional perspective, with asymmetrical contribution, the right fusiform gyrus is known to subserve conscious processing of faces, and the left homolog engages in more general visual perception and object recognition[120,121]. From a clinical perspective, abnormalities in structural asymmetry between the fusiform gyri are related to ASD symptom severity, which often co-occurs with disturbances in speech development[122].

In conclusion, our collection of findings demonstrates that CNVs – an alteration of a human's genetic blueprint present throughout cells of the body and brain – can impact brain morphology in a way that deviates brain asymmetry in several systematic ways. Through functional annotation of the characteristic asymmetry patterns, we demonstrated that CNVs impact preferentially higher-order executive functions. These results might help explain why both CNVs and systematically deviated brain asymmetry are often observed in individuals with autism or schizophrenia. The brain needs to maintain an optimal balance between the size of the left and right regions in the intact central nervous systems[2]. A part of this balance might be mediated through our genetic makeup.

More broadly, we have established how circumscribed, recurring genetic alterations systematically impact one of the pillars of human brain architecture. Adding to earlier observations of structural planum temporale asymmetry *in utero*, we demonstrate that some aspects of the core infrastructure serving uniquely human capacities are engraved in our genetic code. Specifically, our findings on genetically controlled aspects of brain lateralization were highly related to human psychological and behavioral characteristics, such as mapping language and social representational spaces, as well as speech and music comprehension. These insights place regional asymmetry on the (yet to be detailed) causal pathway from gene to brain biology to behavior. Our findings can thus be of great value to all pediatric healthcare institutions. We provide new explanations for observed cognitive and language impairments, while adding a key jigsaw puzzle to the conundrum of gene-phenotype correspondence with consequences for human-defining traits.

## Methods
### Multisite clinical cohort
In the clinically ascertained cohort, signed consent was obtained from all participants or legal representatives prior to inclusion in the study. The current study, which is a secondary data analysis, was approved by the Research Ethics Board (Project 4165) of the Sainte Justine Hospital, Montreal, Canada. UK Biobank participants gave written, informed consent for the study, which was approved by the Research Ethics Committee. The present analyzes were conducted under UK Biobank application number 40980. Further information on the consent procedure can be found online (biobank.ctsu.ox.ac.uk/crystal/field.cgi?id=200).

An extensive description of methods for preprocessing the structural magnetic resonance imaging (MRI) data is available in an already published study with an identical dataset[38]. The participant sample used in this study represents a combination of a carefully collected multi-site clinical cohort and the UK Biobank. Specifically, we grouped 295 CNV carriers identified in the UK Biobank and 257 CNV carriers from the clinical cohorts. In total, our dataset consisted of volumetric asymmetries derived from structural MRI brain scans of 842 subjects: 552 CNV carriers and 290 controls from the clinical ascertainment not carrying any CNV (Table 1).

The here-examined CNVs are among the most commonly studied target CNVs[63]. Specifically, these CNV loci were also selected in other studies, such as an independent research study conducted by The Enhancing NeuroImaging Genetics through Meta-Analysis copy number variant (ENIGMA-CNV) on cognitive, psychiatric, and behavioral manifestations[123]. The 22q11.2, 16p11.2, 1q21.1, and 15q11.2 genomic loci encompass 49, 27, 7, and 4 genes, respectively[124–126]. Analyzed CNV carriers did not carry any other known large CNV. Control individuals did not carry any CNV at these loci. These CNV carriers were either probands referred to the genetic clinic for the investigation of neurodevelopmental and psychiatric disorders or their relatives (parents, siblings, and other relatives).

The identification of CNVs using SNP array (GRCh37/hg19) data followed previously published methods[127,128]. Array quality control was performed based on standard protocols. Using PLINK v1.9[129], we removed SNP variants with a missing rate > 5% as well as SNPs with a Hardy-Weinberg equilibrium exact test $P$-value < 0.0001. We only considered arrays with call rate ≥ 99%, log R ratio SD < 0.35, B allele frequency SD < 0.08, and the absolute value of wave factor <0.05. Furthermore, all individuals with duplicated data or with discordant phenotypic and genetic information regarding sex were removed.

CNVs were called using the pipeline described at https://github.com/labjacquemont/MIND-GENESPARALLELCNV. In short, only CNVs detected by both PennCNV[130] and QuantiSNP[131] were used to minimize the number of potential spurious findings. The resulting CNV calls are available for download from the UK Biobank returned datasets (Return ID: 3104, https://biobank.ndph.ox.ac.uk/ukb/dset.cgi?id=3104). All identified CNVs met stringent quality control criteria: confidence score ≥ 30 (for at least one of the two detection algorithms), size ≥ 50 kb, unambiguous type (deletion or duplication), overlap with segmental duplicates, and HLA regions or centromeric regions <. 50%. Finally, all carriers of a structural variant ≥ 10 Mb, a mosaic CNV, or a chromosome anomaly (aneuploidy or sexual chromosome anomaly) were removed.

Recurrent CNVs at the 4 selected genomic loci were defined based on the following criteria: i) The CNV shows reciprocal overlap > 40% with one of the 4 loci: 16p11.2 proximal (BP4-5, 29.6-30.2MB), 1q21.1 distal (Class I, 146.4–147.5MB & II, 145.3–147.5MB), 22q11.2 proximal (BPA-D, 18.8–21.7MB) and 15q11.2 (BP1-2, 22.8–23.0MB). ii) CNV includes all of the coding genes within one of the 4 unique genomic loci. As a result, recurrent CNVs at a given loci are 100% identical with respect to coding regions. These CNVs were visually inspected by at least two researchers to confirm these criteria.

## Clinical MRI data: recording, quality control and pre-processing protocols

We analyzed a subject sample with T1-weighted (T1w) brain images at 0.8–1 mm isotropic resolution. All T1w included in the analysis were quality-checked by a domain expert[63]. Data for Voxel-Based Morphometry were preprocessed and analyzed with SPM12[132–134] running under MATLAB R2018b (https://www.mathworks.com/products/new_products/release2018b.html). Further quality control was performed using standardized ENIGMA quality control procedures (http://enigma.ini.usc.edu/protocols/imaging-protocols/). All MRI scans had complete brain coverage, including the cerebellum. Finally, neurobiologically interpretable measures of gray matter volume were extracted in all participants by summarizing whole-brain MRI maps in the MNI reference space by means of anatomical reference atlases. This feature-generation step was guided by the topographical brain region definitions based on the combination of hemisphere-symmetrical Harvard-Oxford and the 2009 Diedrichsen atlas (http://fsl.fmrib.ox.ac.uk/fsl/fslwiki/Atlases)[135]. For each participant, we thus derived image-derived quantities of a total of 130 local gray matter volumes across the whole brain: 96 cortical, 14 subcortical, and 20 cerebellar volume measures. All subjects had complete coverage of all regions under study.

## Population MRI data: UK Biobank data source

The UK Biobank is the largest existing biomedical dataset. It offers extensive behavioral and demographic assessments, medical and cognitive measures, as well as biological samples in a cohort of ~500,000 participants recruited from across Great Britain (https://www.ukbiobank.ac.uk/). The present study was based on the brain-imaging data released from February/March 2020. Our population sample included measurements from 36,742 participants with brain-imaging measures and expert-curated image-derived phenotypes of gray matter morphology (T1-weighted MRI). We further subselected only those participants who carry one of the eight here-studied CNVs. Among the 302 identified CNV carriers, 141 were men, and 161 were women aged between 40 and 69 years old when recruited (mean age: 54 years old, standard deviation 7 years). We benefited from the uniform data preprocessing pipelines designed and implemented by the FMRIB team, Oxford University, Oxford, UK[136] to improve comparability and reproducibility to other studies using this population dataset.

MRI scanners (3 T Siemens Skyra) at several dedicated data collection sites used matching acquisition protocols and standard Siemens 32-channel radiofrequency receiver head coils. Brain imaging scans were defaced to protect the study participants' anonymity, and any sensitive meta-information was removed. Automated processing and quality control pipelines were deployed[136,137]. Specifically, the noise was removed using 190 sensitivity features to improve the homogeneity of the brain imaging scans. This approach allowed for the reliable identification and exclusion of problematic brain scans, such as due to excessive head motion.

The structural brain MRI data were acquired as high-resolution T1-weighted images of brain anatomy using a 3D MPRAGE sequence at 1 mm isotropic resolution. It was preprocessed including gradient distortion correction, the field of view reduction using the Brain Extraction Tool[138] and FLIRT[139], as well as non-linear registration to MNI152 standard space at 1 mm resolution using FNIRT[140]. All image transformations were estimated, combined, and applied in a single step to avoid unnecessary interpolation. Tissue-type segmentation into the cerebrospinal fluid, gray matter, and white matter to generate full bias-field-corrected images was achieved using FAST (FMRIB's Automated Segmentation Tool[141]). Finally, gray matter images were used to extract gray matter volumes in parcels according to the combination of the Harvard-Oxford and the 2009 Diedrichsen atlas (http://fsl.fmrib.ox.ac.uk/fsl/fslwiki/Atlases)[135]. Specifically, we extracted the 130 gray matter volumes across the whole brain from category 1101: Regional gray matter volumes (FAST).

## Asymmetry index to quantify regional left-right divergence

All subsequent analyzes were performed in Python v3.8 as scientific computing engine (https://www.python.org/downloads/release/python-380/). For each of our subjects from the UK Biobank and clinical sample, we had 65 homolog contralateral volume measures (i.e., 130 measurements in total across the brain, cf. above). More specifically, within the 65 pairs of regions, each subject was characterized by 48 cortical, 7 subcortical, and 10 cerebellar volume measures in each hemisphere. As a data-cleaning step, all derived regional brain volumes were adjusted for variation that can be explained by the scanning site (Sup. Fig. 5). We did not adjust for the effects of intracranial volume, age, or sex, as these did not affect subsequent brain asymmetry patterns (Sup. Fig. 6). Finally, for each pair of homologous contralateral regions, we calculated asymmetry indices using the left (L) and right (R) regional volumes as $(L - R) / ((L + R) / 2)$[142]. Therefore, a positive AI signifies leftward asymmetry,

while a negative AI represents hemispheric features whose volume skews dominantly rightwards.

## Multivariate asymmetry pattern learning pipeline

Technically, our core aim was to derive robust brain asymmetry patterns that separate between CNV carriers and controls using the 65 asymmetry indices characterizing each subject's 130 atlas volumes of paired regions. We derived the asymmetry patterns as systematic brain morphometric co-deviations attributable to each of our eight target CNVs. To this end, we capitalized on linear discriminant analysis to extract separating rules between CNV carriers and controls based on asymmetry indices[38]. LDA can be viewed as a generative approach to separating CNV carriers from controls, which requires fitting multivariate Gaussian distribution to regional brain asymmetries[143]. To do so, LDA has access to class labels (CNV status vs control in our case) and thus belongs to a supervised category of pattern-learning algorithm techniques[143]. Using a linear model represents a data-efficient and directly biologically interpretable approach to our analysis, which is ideally suited to our data analysis scenario, especially given the sample size of clinical subject samples such as ours[144]. Clinical datasets are characteristic of the low sample size regularly encountered in biology and medicine, which typically impedes the application of more complex non-linear models that require high numbers of parameters and/or non-linear relationships to be estimated[145].

In our study, we brought to bear LDA models to separate between CNV carriers and controls based on asymmetry measures derived from brain atlas region volumes. These asymmetry measures were z-scored to zero mean and unit-variance scaling across relevant participants. Specifically, we derived a single LDA prototype for each CNV type, which yielded eight CNV-specific models. After estimating these dedicated LDA models, for the sake of neurobiological interpretation, we inspected which regional asymmetries were the most informative in telling apart CNV carriers and controls. In other words, we aimed to contextualize and explicitly unpack the discriminatory principles of our LDA models. Each LDA model estimated a morphological asymmetry representation by a set of 65 coefficients pertaining to 65 regional asymmetries (i.e., 48 for cortical, 7 for subcortical, and 10 for cerebellar homologue regions). This set of estimated coefficient values revealed the concomitant contribution of each brain region asymmetry towards the separability of the CNV carriers based on whole-brain asymmetry measurements. Therefore, the coefficients provided explicit quantitative information on the relative importance of the collective asymmetries for CNV-health separation. Moreover, the derived LDA coefficients were estimated hand-in-hand with the other brain region asymmetry effects, in contrast to the estimation of marginal or partial variable effects as in linear regression[146].

## Performing model inspection using bootstrap resampling

As an acid test for robust brain region effects, we adopted a criterion to test which LDA coefficients designated robust above-chance brain effects, meaning which asymmetry indices significantly contribute to the separation. For that purpose, we embraced a bootstrapping resampling strategy for the LDA models separately for all eight CNV classes. Specifically, we used the following analytical protocol for a set of subjects consisting of a single CNV type and controls. In the first phase, a randomly perturbed companion version of the original dataset was created by sampling a subject cohort with the same sample size with replacement. We repeated the bootstrap resampling procedure with 1000 iterations and executed the LDA approach on each of the ensuing alternative dataset instances. In so doing, we obtained different realizations of the entire analysis workflow and ensuing LDA model estimates. Concretely, the bootstrapping algorithm resulted in 1000 instances of the trained LDA models used to obtain 1000 sets of associated LDA coefficients.

Since the 1000 LDA model instances were derived from different bootstrap draws of the original cohort, it could happen that two distinct LDA models' coefficients would carry opposite signs due to the sign invariance of LDA dimensionality reduction. To address this form of reflection invariance, we ensured alignment of all LDA models by multiplying them with −1 or 1 to produce a positive correlation between LDA coefficients and a corresponding Cohen's $d$ map. This Cohen's $d$ map is calculated as Cohen's $d$ between respective CNV carriers and controls independently for all 65 regions.

Finally, statistically salient coefficients had a distribution of 1000 LDA coefficients robustly different from 0. Specifically, they were interpreted as robustly different from zero if their two-sided confidence interval according to the 2.5/97.5% bootstrap-derived distribution did not include zero coefficient value, indicating the absence of an effect[147]. Finally, we compared LDA coefficients between different CNVs using Pearson's correlation. We used spin permutation testing to calculate empirical $P$-values for the ensuing correlation coefficient[148].

## Building multi-class prediction models

As another key model property of direct relevance to our present analysis goals, LDA can also be viewed as a dimensionality technique. Specifically, LDA inherently projects the input subjects' set of brain morphology measurements into a linear subspace consisting of the directions which maximally separate our classes[149]. As a general rule, the maximum number of latent factor dimensions equals the number of classes −1. Therefore, the LDA model instance built to discriminate between eight classes at hand (e.g., eight different CNVs) can be characterized by a seven-dimensional vector of associated LDA coefficients. We used the first two dimensions with the highest discriminative capacity to identify regions contributing to the separation between CNVs.

## Descriptive statistical analysis for volumetric brain measures

To complement the analyses underlying the impact of CNVs on brain asymmetry, we investigated how CNVs impact the raw regional volumes in each hemisphere separately. These ancillary analyses into the CNV effects on the left and right hemispheres used the original 130 regional brain volumes adjusted for intracranial volume, age, $age^2$, sex, and scanning site, following previous research on this dataset[63]. We used Cohen's $d$ to quantify the effect size of the CNVs on individual regional volumes. For a given region, Cohen's $d$ is defined as:

$$d = \frac{\bar{x_1} - \bar{x_2}}{\sqrt{\frac{s_1^2 + s_2^2}{2}}},$$

where $\bar{x_1}$ corresponds to the mean region volume across CNV carriers, $\bar{x_2}$ corresponds to the mean region volume across controls. Similarly, $s_1$ and $s_2$ correspond to standard deviations of CNV carriers and controls.

In addition, we computed a bootstrap confidence interval for all Cohen's $d$ values to get an estimate of the uncertainty associated with the calculated effect size. Specifically, we calculated Cohen's d for each bootstrap resampled dataset of CNV carriers and controls. The bootstrap resampling involved repeated sampling with replacement from the original regional volumes separately from CNV carriers and controls. This process was repeated 1000 times to obtain a confidence interval based on 2.5% and 97.5% percentile of resampled Cohen's $d$ value distribution.

## Functional annotation profiling via NeuroSynth

We capitalized on the NeuroSynth resource to provide a data-driven functional characterization for the derived CNV-specific brain asymmetry patterns. NeuroSynth represents one of the largest platforms for large-scale synthesis of functional MRI data available to date[49]. It uses data-mining techniques to search for concepts of interest in the

neuroimaging literature. Specifically, NeuroSynth associates activation coordinates with high-frequency keywords in each study from the database. There were 507,891 activation peaks reported in 14,371 studies when we queried the database in December 2022. Next, we correlated all 3228 term-specific activation maps with the brain asymmetry patterns pertinent to 16p11.2 and 22q11.2 deletions. We used the absolute values of asymmetry patterns for bilateral brain regions to define a single mask encompassing the whole brain in standard space (Montreal Neurological Institute space 152). Obtained correlations highlight the spatial overlap between functional activations associated with a given keyword and the CNV-specific brain asymmetry pattern.

We complemented our NeuroSynth exploration by associating signed brain patterns (compared to the absolute described above) with more specific functional descriptions following the methodology from previous studies[150,151]. Specifically, we associated the spectrum of LDA coefficients with a list of 24 predefined cognitive domain terms[150]. To investigate how the positive and negative LDA coefficients decode functions separately, the original LDA coefficients were used for the right-hemisphere regions. LDA coefficients with the opposite sign were used for the left-hemisphere regions. Negative LDA coefficients represented decreased rightward or increased leftward asymmetry and vice versa for positive coefficients. In the next step, we projected the coefficient of bilateral regions to the whole cortex. Finally, we binned ranked voxels into five-percentile increments along the coefficient spectrum. Similarly, we binned each ranked term-specific activation map. Binned activation maps were z-scored, and bins with a z-score lower than 0.5 were set to 0. Finally, we calculated a weighted score as a product of binned activation and binned LDA coefficients. The obtained score served as a proxy to assess how much the function terms were associated with positive and negative LDA coefficients.

### Genotyping, imputation, and quality control
The UK Biobank release contained both genotype and brain-imaging data for all individuals. Imputations were performed against the Haplotype Reference Consortium (HRC), UK10K haplotype resource, and 1000 Genomes Phase 3 reference panels. Specific detail of the genotyping quality control was described elsewhere[152]. In addition to the quality control performed by the UK Biobank, we followed more stringent quality control metrics previously used by the Neale Lab (http://www.nealelab.is/uk-biobank/). We further excluded participants who i) had a mismatch between genetically inferred sex and self-reported sex, ii) had high genotype missingness or extreme heterozygosity, and iii) were excluded from the kinship inference process or had ten or more 3rd-degree related relatives identified. To acknowledge ethnicity as a major source of population stratification, we only included individuals of European descent in our analyzes. To this end, we restricted our analysis to individuals who self-identified by questionnaire as 'White', 'British', 'Irish', or 'Any other white background' and whose samples were used to compute the genetic principal components (data field 22020). Participants who were not within 7 standard deviations for the leading 6 PCs were excluded from further analysis (http://www.nealelab.is/uk-biobank/). Using PLINK, we filtered out subjects who had a high missing genotype rate ( > 1%) and genetic variants that had a high missing rate ( > 1%), low minor allele frequency ( < 0.001), low imputation INFO score ( > 0.8), and significant deviation from Hardy-Weinberg equilibrium ($P$-value < $1 \times 10^{-10}$) (http://www.nealelab.is/uk-biobank/). After participant and genotyping quality control, we restricted the analysis to a set of 29,470 participants of white-British ancestry.

### GWAS analyzes of planum temporale
Following careful curation of the phenotypic and genotypic data, the genome-wide association analysis of the brain asymmetry, volume left, and volume right phenotypes in the UKB data was conducted using the fastGWA tool implemented in the Genetic Complex Trait Analysis (GCTA) software[153]. The fastGWA is a widely used tool for genome-wide analyzes of biobank-scale data that controls for population stratification and for relatedness[154]. To correct for potential subtle population stratification effects, we included the first 20 UKB-provided genetic principal components as covariates. Following the recommendation of the developers and a previously implemented association model that ran the GWAS analysis of 7,221 phenotypes in the UK Biobank (https://github.com/Nealelab/UK_Biobank_GWAS), additional covariates in the analysis were: age, sex, age2, sex*age, sex*age[2]. As a sensitivity analysis, we tried including total brain volume as an additional covariate, but it did not influence the presented results (Sup. Fig. 7).

As the next step, we estimated genetic correlations between the three measurements of planum temporale, ASD, and SCZ. GWAS summary statistics for ASD and SCZ were obtained from the Psychiatric Genomics Consortium (PGC). Both resources represent the currently largest GWAS analysis on the respective disorder[155,156]. Calculating the genetic correlations was done using linkage disequilibrium (LD) score regression (LDSC, v.1.0.0)[157], which is based on GWAS summary statistics generated in this study and does not require individual-level data.

### Functional mapping and annotation of GWAS via FUMA
FUMA builds on Multi-marker Analysis of GenoMic Annotation (MAGMA), a gene analysis tool for GWAS data used in our downstream analyzes, to annotate input summary statistics by mapping lead SNPs to genes. FUMA first takes GWAS summary statistics as an input and provides extensive functional annotation for all SNPs in genomic areas identified by lead SNPs. Then, FUMA identifies a list of gene IDs from the lead SNPs[158]. In addition, we used this platform to visualize obtained results using Manhattan plots, QQ plots, and regional association plots.

### Reporting summary
Further information on research design is available in the Nature Portfolio Reporting Summary linked to this article.

## Data availability
The majority of 16p11.2 data are publicly available (https://www.sfari.org/). For the 22q11.2 sample, raw data are available upon request from the PI (CB). All derived measures used in this study are available upon request (SJ). The rest of the CNV carriers' data cannot be shared as participants did not provide consent. All data from UK Biobank are available to other investigators online (ukbiobank.ac.uk). The Harvard-Oxford and Diedrichsen atlases are accessible online (http://fsl.fmrib.ox.ac.uk/fsl/fslwiki/Atlases). Source data are provided as a Source Data file. Source data are provided with this paper.

## Code availability
The processing scripts and custom analysis software used in this work are available in a publicly accessible GitHub repository, along with examples of key visualizations in the paper: https://github.com/dblabs-mcgill-mila/CNV-asymmetry.

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

## Acknowledgements

DB was supported by the Brain Canada Foundation, through the Canada Brain Research Fund, with the financial support of Health Canada, National Institutes of Health (NIH R01 AG068563A, NIH R01 DA053301-01A1, NIH R01 MH129858-01A1), the Canadian Institute of Health Research (CIHR 438531, CIHR 470425), the Healthy Brains Healthy Lives initiative (Canada First Research Excellence fund), Google (Research Award, Teaching Award), and by the CIFAR Artificial Intelligence Chairs program (Canada Institute for Advanced Research). This research was supported by Calcul Quebec (http://www.calculquebec.ca) and Compute Canada (http://www.computecanada.ca), the Brain Canada Multi-Investigator initiative, the Canadian Institutes of Health Research, CIHR_400528, The Institute of Data Valorization (IVADO) through the Canada First Research Excellence Fund, Healthy Brains for Healthy Lives through the Canada First Research Excellence Fund. SJ is a recipient of a Canada Research Chair in neurodevelopmental disorders and a chair from the Jeanne et Jean Louis Levesque Foundation. The Cardiff CNV cohort was supported by the Wellcome Trust Strategic Award "DEFINE" and the National Centre for Mental Health with funds from Health and Care Research Wales (code 100202/Z/12/Z). The CHUV cohort was supported by the SNF (Maillard Anne, Project, PMPDP3 171331). Data from the UCLA cohort provided by CEB (participants with 22q11.2 deletions or duplications and controls) was supported through grants from the NIH (U54EB020403), NIMH (R01MH085953, R01MH100900, R03MH105808), and the Simons Foundation (SFARI Explorer Award). KK was supported by The Institute of Data Valorization (IVADO) Postdoctoral Fellowship program through the Canada First Research Excellence Fund. IES is supported by the Research Council of Norway (#223273), South-Eastern Norway Regional Health Authority (#2020060), European Union's Horizon2020 Research and Innovation Programme (CoMorMent project; Grant #847776) and Kristian Gerhard Jebsen Stiftelsen (SKGJ-MED-021). We thank all families participating at the Simons Searchlight sites and 16p11.2 European Consortium. We appreciate obtaining access to brain-imaging and phenotypic data on SFARI Base. We are grateful to all families who participated in the 16p11.2 European Consortium. The funders had no role in study design, data collection and analysis, the decision to publish or the preparation of the manuscript.

## Author contributions

D.B., J.K., and S.J. designed the study, analyzed imaging data, and drafted the manuscript. J.K., C.Mod., and K.K. did all the preprocessing and analysis of neuroimaging data. KSh performed GWAS and genetic correlation analyzes. D.B. and S.J. contributed to the interpretation of the results and in the editing of the manuscript. C.Mod. and B.D. recruited and scanned participants in the 16p11.2 European Consortium. S.L., C.O.M., N.Y., and E.D. recruited and scanned participants in the Brain Canada cohort. L.K. and C.E.B. collected and provided the data for the UCLA cohort. D.E.J.L., M.J.O., M.B.M.Vd.B., J.H., and A.I.S. provided the data for the Cardiff cohort. K.K., K.Sh., K.Sa., C.Mod., C.Mor., G.H., M.J.L., C.O.M., Z.S., N.Y., E.D., K.J., A.B.C., L.K., A.I.S., M.B.M.Vd.B., D.E.J.L., M.J.O., J.H., S.L., B.D., I.E.S., O.A.A., D.C.G., P.M.T., C.E.B., and R.Z. provided feedback on the manuscript. D.B. led data analytics.

## Competing interests

DB is a shareholder and advisory board member of MindState Design Labs, USA. OAA is a consultant to Cortechs.ai. PT obtained grant support

from Biogen, Inc., for research unrelated to this manuscript. The remaining authors declare no competing interests.

## Additional information

[1]Mila - Québec Artificial Intelligence Institute, Montréal, QC, Canada. [2]Department of Biomedical Engineering, Faculty of Medicine, McGill University, Montreal, Canada. [3]Centre de recherche CHU Sainte-Justine, Montréal, Quebec, Canada. [4]LREN - Department of Clinical Neurosciences, Centre Hospitalier Universitaire Vaudois and University of Lausanne, Lausanne, Switzerland. [5]Imaging Genetics Center, Stevens Neuroimaging and Informatics Institute, Keck School of Medicine of USC, Marina del Rey, CA, USA. [6]Institut universitaire en santé mentale de Montréal, University of Montréal, Montréal, Canada. [7]Department of Psychiatry, University of Montreal, Montréal, Canada. [8]Semel Institute for Neuroscience and Human Behavior, Departments of Psychiatry and Biobehavioral Sciences and Psychology, UCLA, Los Angeles, USA. [9]School for Mental Health and Neuroscience, Maastricht University, Maastricht, Netherlands. [10]Centre for Neuropsychiatric Genetics and Genomics, Cardiff University, Cardiff, UK. [11]Division of Psychological Medicine and Clinical Neurosciences, School of Medicine, Cardiff University, Cardiff, UK. [12]Neuroscience and Mental Health Innovation Institute, Cardiff University, Cardiff, UK. [13]Neurology Department, Max-Planck-Institute for Human Cognitive and Brain Sciences, Leipzig, Germany. [14]NORMENT, Division of Mental Health and Addiction, Oslo University Hospital and University of Oslo, Oslo, Norway. [15]Department of Medical Genetics, Oslo University Hospital, Oslo, Norway. [16]KG Jebsen Centre for Neurodevelopmental Disorders, University of Oslo, Oslo, Norway. [17]Department of Psychiatry, Boston Children's Hospital and Harvard Medical School, Boston, MA, USA. [18]International Laboratory for Brain, Music and Sound Research, Montreal, QC, Canada. [19]Department of Pediatrics, University of Montréal, Montréal, Quebec, Canada. [20]TheNeuro - Montreal Neurological Institute (MNI), McConnell Brain Imaging Centre, Faculty of Medicine, McGill University, Montreal, QC, Canada. ✉e-mail: danilo.bzdok@mcgill.ca

