## [Peer Review File · Nature Communications]

Using rare genetic mutations to revisit structural brain asymmetryREVIEWER COMMENTS

Reviewer #1 (Remarks to the Author):

summary

This is a very interesting study that sheds further light into the relevance of structural asymmetries for key human traits, taking well-characterized CNVs as a starting points. The results highlighted the planum temporale (PT) as a key region that is most affected across several CNV carriers, which was further characterized genetically by common genetic variation associated to PT asymmetry (+ left and right) using the UK biobank.

major comments

-

- For someone with background in genetics and brain asymmetry, but not so much knowledge of the CNVs, I got the feeling that there is not enough exploitation of the information that the specific CNVs give in these analyses. I would have appreciated a bit more interpretation on how each CNV affects different phenotypes (i.e. related to language or other, there is only a mention to CAS and speech-language disorder in the introduction, but no much further information given in the results/discussion.

- (related) the rationale to present first only the analyses of 2 CNVs and then expanding it to the 8 CNVs is unclear, and it also makes the results of the 8 CNVs a bit redundant with the first. I would maybe present first the complete analysis, and then zoom into those two specific CNVs if needed, explaining why they are of particular interest.

- Expanding the genetic correlation analysis to include r_g 's with ASD and SCZ could be helpful to either confirm previously found null r_g , or to understand how the PT asymmetry genetically relates to these phenotypes both at the rare CNV level, as well as the level of common genetic variation.

- I think the discussion section needs some improvement. Part of it covers a wide range of information about the planum temporale-associated phenotypes, without putting this in the context of the current study (and its starting point from the CNV analysis). I found some of

its parts a bit farfetched (e.g. the role of music in society) for the current study.

minor comments

introduction

- page 4, last paragraph. What is the motivation to focus on 16p11.2 and 22q11.2 deletions to start with? either here or at the beginning of the results, it would be helpful if you guided the reader as to why these two CNVs have been analyzed separately in the first place.

results

- page 5 "Brain asymmetry effects induced by 16p11.2 and 22q11.2 deletions": to which extent are the LDA models good fit? i.e. sensitivity and specificity are not reported.

- pages 5-6 "Direction of magnitude of induced asymmetry indices": for the planum temporale asymmetry, 16p11.2 and 22q11.2 deletion carriers showed an increased leftward asymmetry compared to controls. This is somehow counterintuitive, as the leftward lateralization in the general population is usually interpreted to the leftward asymmetry of language function? Given the language deficit phenotypes these deletion carriers present, I would have expected an opposite pattern (i.e. less lateralized PT? or rightwards?).

Nonetheless, structural PT asymmetry cannot be interpreted as an index of individual variability in language dominance (e.g. see Tzourio-Mazoyer et al. 2018, DOI: 10.1007/s00429-017-1551-7). May be worth discussing further in the discussion.

- Related, from figure 2b and Cohen's d's reported in page 6 (first paragraph) it seems the CNV carriers have larger left *and* right PTs, maybe reflecting a more global effect. I.e. larger brains tend to have larger asymmetries? (see Williams et al. 2022 for a discussion, DOI: 10.1016/j.neuroimage.2022.119118). You have already performed a sensitivity analysis to assess the effect of deconfounding additional factors, and shown that the loadings are highly correlated (Supplementary Figure 3), it may however help understand where the effects are coming from (i.e. maybe they affect differently left and right?).

- page 6, Figure 3a-c: given that the LDA coefficients for 22q11.2 del are a subset of the 16p11.2, I don't find it surprising that there is a high overlap in the functionally associated terms for these regions.

- page 7 "Comparison of asymmetry patterns across eight CNVs" it would be helpful to know how well these LDA models are.

- page 7, when presenting the role of PT and fusiform asymmetries across the 8 CNVs, it is now misleading that they contributed 3/8 CNVs. It would be more fair to say that they contributed to one additional CNV apart from the deletions of 16p and 22q (which you have already presented in the sections above).
- Figure 4b, suggestion: may be worth having a boxplot rather than barplot to show the spread of the LDA coefficients across the 8 CNVs. Maybe also labelling the points that were had significant loadings.
- page 7, lines 256-257: "CNVs known to have deleterious consequences on language performance"... how is 15q11.2 related to language?
- page 8, second paragraph: I'm not entirely sure, but would it make sense to perform a clustering analysis based on the LDA weights / correlation in Figure 4d? If possible, this could inform about how the different CNVs cluster with regard to brain asymmetry, and whether there is a unique or multiple clusters.
- page 8: gene names should be in italics.
- page 8: the SNP associated with PT asymmetry (rs41298373) is coding non-synonymous SNP. It also has a particular LD structure around this hit, reflected by the lack of other SNPs in this locus associated with PT asymmetry (as shown in the Manhattan plot in Figure 5a)
- page 9, lines 311-316: it is not clear to me what do the left and right PT GWAS signal add with regard to the loci that are not significant in the AI GWAS. You mention (lines 323-324) that the CNV effect is driven by a larger left PT, but do not provide any interpretation as to how these signals could contribute to the observed effects.
- page9, third paragraph: the genetic correlation between left and right PT you report is exactly the same as the one in Carrion-Castillo et al. (2020, DOI: 10.1016/j.cortex.2019.11.006).
- The much larger UK biobank sample size that you have used in the current analysis (N ~30,000 vs N~18,000) could be used to explore whether PT AI (or left PT) are genetically correlated with ASD and SCZ (the previous study tested this and found no evidence, but your current GWAS would be better powered). If such a genetic correlation were found, it would be a nice way of linking back to the CNV phenotypes as well.

discussion

- this section reads as a long part about the planum temporale followed by some context for

the other identified asymmetries. I think some more context about the CNVs would be helpful to guide the reader put in

- page 10, lines 357-358: "(...) which demonstrated that the genetic basis of left and right planum temporale volume is partly diverging". This is not new, as it was already explored in detail by Carrion-Castillo et al. (2020, DOI: 10.1016/j.cortex.2019.11.006). Please rephrase, e.g. replace "demonstrated" with "confirmed" or similar.
- page 11, first paragraph: I don't see that you need all this information about music, its role as "social glue" etc. It may be relevant, given the phenotypes of CNV carriers, but no information in this respect is provided.
- page 12, lines 458-460: "Notably..." the association of rs41298373 with left PT but not right PT was already reported by Carrion-Castillo et al. (2020).

methods

- page 16, line 632: why are there 290 controls included as opposed to 1296 controls included in Modenato et al. 2021? please specify how the subset for the current study has been defined.
- page 16, paragraph 3 (starting line 633): I found this paragraph useful, and would have appreciated it in the introduction to justify why these CNVs are of interest.
- page 18: were the LDA models somehow evaluated?

Table 1

- page 24: Table 1 caption. When defining the ICD10 codes that correspond to each diagnosis labels are flipped: line 927 should read "SCZ" instead of "ASD", line 929 should read "ASD" instead of "SCZ".
- page 24: Table 1 caption. Why are the UKB diagnosis definitions included here? If part of the dataset is from the UKB, please specify how many. For the non-UKB samples, do they also have diagnosis information? Also, given the age differences between CNV carriers and controls, specify age range in addition to sd?

Reviewer #2 (Remarks to the Author):

Review of Nature Communications Manuscript NCOMMS-23-22437-T

In their manuscript entitled “Using rare genetic mutations to revisit structural brain asymmetry” Kopal and co-workers used UK Biobank data and other data to investigate the role of rare genetic mutations in structural hemispheric asymmetries in the human brain. The most noteworthy result is that asymmetry of the planum temporale, a well-known brain area relevant for language processing was especially susceptible to deletions and duplications of specific gene sets. I think that the work could be of significance to neuroscience and related fields. The established literature on hemispheric asymmetries mostly focused on common variants, with few exceptions. Moreover, the established literature focused mostly on functional hemispheric asymmetries, not structural hemispheric asymmetries, as in this work. The work supports the conclusions and claims, and I do not see any flaws in the data analysis, interpretation, and conclusions that would prohibit publication. The methodology is sound, and the work meets the expected standards in the field.

In general, I could see it published in NCOMMS, but have a few comments that the authors may wish to address in a revised version of the manuscript.

For now, I recommend:

Minor revisions

Specific points:

General:

Please check whether the font size is uniform throughout the manuscript. There were some somewhat irritating changes in font size in the manuscript (e.g., between the first and the second sentence in the abstract).

Introduction

Statement “Enabled by a multicohort machine learning approach, we quantitatively dissected the impact on brain asymmetry of eight CNVs: deletions and duplications at 1q21.1 distal, 15q11.2 BP1-BP2, 16p11.2 proximal, and 22q11.2 proximal loci, with a particular focus on deletions at 16p11.2 and 22q11.2 loci as the two most studied CNVs with high penetrance to neuropsychiatric disorders.”

The readers may not be immediately familiar with these specific CNVs. It would be great if

the authors could provide more information on them such as likely associated genes and likely associated functions as well as references to the published literature that may relate them to asymmetries. In general, the authors should explain a bit more why these specific CNVs were chosen for studying brain asymmetries.

I am not sure about the journal's policy on citing preprints that have not undergone peer review, but the following preprint focused on rare variants in handedness, a form of functional hemispheric asymmetries that may be related to structural hemispheric asymmetries may be worthwhile to include in the intro:

<https://www.biorxiv.org/content/10.1101/2023.05.31.543042v1>

Moreover, the following study also used whole exome sequencing in functional hemispheric asymmetries and may be relevant:

Kavaklioglu T, Ajmal M, Hameed A, Francks C. Whole exome sequencing for handedness in a large and highly consanguineous family. *Neuropsychologia*. 2016 Dec;93(Pt B):342-349.

It would be great if the authors could present a hypothesis at the end of the introduction. While I understand that this is largely a data-driven study, there must have been something that pointed the authors into the general direction of conducting this study.

Results

The UK Biobank data contain information on human handedness and previous studies on structural hemispheric asymmetries in the UK Biobank data have made links between structural asymmetries and handedness, e.g.:

Sha Z, Pepe A, Schijven D, Carrión-Castillo A, Roe JM, Westerhausen R, Joliot M, Fisher SE, Crivello F, Francks C. Handedness and its genetic influences are associated with structural asymmetries of the cerebral cortex in 31,864 individuals. *Proc Natl Acad Sci U S A*. 2021 Nov 23;118(47):e2113095118.

It would be highly interesting and relevant if the authors could see how the findings are linked to handedness.

Discussion

“Hence, in neurotypical individuals, the planum temporale is highly structurally lateralized, with 65% of all individuals showing larger left planum temporale; and only 11% of individuals showing larger right planum temporale volume”

There is a longstanding discussion on whether the macrostructural asymmetry of the planum is relevant for function or whether microstructural asymmetries are more relevant. These should therefore also be mentioned as they may also be affected by the genetic variation found and the discussion should be a bit more balanced in this regard.

The following paper may be relevant:

Galuske RA, Schlote W, Bratzke H, Singer W. Interhemispheric asymmetries of the modular structure in human temporal cortex. *Science*. 2000 Sep 15;289(5486):1946-9. doi: 10.1126/science.289.5486.1946

Ocklenburg S, Friedrich P, Fraenz C, Schlüter C, Beste C, Güntürkün O, Genç E. Neurite architecture of the planum temporale predicts neurophysiological processing of auditory speech. *Sci Adv*. 2018 Jul 11;4(7):eaar6830. doi: 10.1126/sciadv.aar6830

Methods

Statement: “UK Biobank participants gave written, informed consent for the study, which was approved by the Research Ethics Committee.”

Please also include ethics information for participants that were not from the UK Biobank.

Asymmetry index:

Please give reference and some more info on why $(L - R) / ((L + R) / 2)$ was used and not the more common $(L - R) / (L + R)$.

NCOMMS-23-22437-T

Newly added text is marked in purple: Example

Deleted text is strikethrough and red: ~~Example~~

Full set of edits is available in the attached document with tracked changes.

Reviewer #1 (Remarks to the Author)

summary

This is a very interesting study that sheds further light into the relevance of structural asymmetries for key human traits, taking well-characterized CNVs as a starting points.

The results highlighted the planum temporale (PT) as a key region that is most affected across several CNV carriers, which was further characterized genetically by common genetic variation associated to PT asymmetry (+ left and right) using the UK biobank.

We are grateful to the reviewer for the positive assessment of our work.

major comments

- For someone with background in genetics and brain asymmetry, but not so much knowledge of the CNVs, I got the feeling that there is not enough exploitation of the information that the specific CNVs give in these analyses. I would have appreciated a bit more interpretation on how each CNV affects different phenotypes (i.e. related to language or other, there is only a mention to CAS and speech-language disorder in the introduction, but no much further information given in the results/discussion.

We thank the reviewer for highlighting this important point. Single CNV analyses with detailed descriptions of their effects have been done in previous papers. Here, we adopted an across-CNV angle, which is novel. We have now expanded the description of CNVs, and we make the novel across-CNV approach clear in the Introduction.

Intro:

CNVs at the 22q11.2, 16p11.2, 1q21.1, and 15q11.2 genomic loci are among the most commonly identified risk factors for neuropsychiatric disorders in pediatric clinics (Moreno-De-Luca 2013; Crawford et al., 2019). These deletions and duplications of genomic sequences strike a balance between occurrence in the population and their strong biological consequences. In other words, while these CNVs are rare, especially compared to single nucleotide polymorphisms studied in GWAS, these genetic alterations are frequent enough (between 1 in 500 and 1 in 4000 in the general population) so that we can begin to carry out across-CNV investigations in population datasets.

Specifically, these CNVs are now being understood to exert body-wide implications, including the cardiovascular, endocrine, skeletal and nervous systems (Auwerx et al., 2022; Kopal et al., 2023), with deteriorating consequences to everyday life (Kendall et al., 2021). The substantial downstream effects suggest that CNVs may serve as a sharp imaging-genetics tool for interrogating the effects of genetic alterations on brain physicality and behavioral differentiation³⁵. ~~Studies have already started to delineate the effects of CNVs on brain structure^{36,37} and function³⁸⁻⁴⁰.~~ Although CNVs are well known to impact some of the most lateralized cognitive functions, including language skills⁴¹, how CNVs affect structural brain asymmetry has not yet been explored. 16p11.2 and 22q11.2 deletions stand out as the two “relatively” frequent CNVs with large effects on risk for neuropsychiatric disorders⁴⁵. The most substantial increase in risk for schizophrenia (SZ) is associated with 22q11.2 deletions (30 to 40-fold increase) (Marshall et al., 2017), while 16p11.2 deletions confer notably elevated risk (10-fold increase) for autism spectrum disorder (ASD) (Sanders et al., 2015). In terms of language impairments, ~~To illustrate the apparent CNV-induced language impairments,~~ 77% of children and 50% of adults carrying a 16p11.2 deletion meet the criteria for childhood apraxia of speech⁴². Furthermore, 95% of children with a 22q11.2 deletion are diagnosed with speech-language disorders⁴³. Speech and language delays are hallmark features of CNVs in pediatric clinics; perhaps the earliest symptoms for which children with a CNV get clinically referred in the first place⁴⁴.

In this study, we have leveraged **multiple** rare high-effect-size deletions and duplications to investigate how gene dosage modulates brain asymmetry **as well as to examine the influence of genetic loci**. Enabled by a multi-cohort machine learning approach, we quantitatively dissected the impact on brain asymmetry **across** eight CNVs: deletions and duplications at 1q21.1 distal, 15q11.2 BP1-BP2, 16p11.2 proximal, and 22q11.2 proximal loci, with a particular focus on deletions at 16p11.2 and 22q11.2 loci. ~~as the two most studied CNVs with high penetrance to neuropsychiatric disorders⁴⁵~~.

- (related) the rationale to present first only the analyses of 2 CNVs and then expanding it to the 8 CNVs is unclear, and it also makes the results of the 8 CNVs a bit redundant with the first. I would maybe present first the complete analysis, and then zoom into those two specific CNVs if needed, explaining why they are of particular interest.

We thank the reviewer for the valuable suggestion. We carefully revised the whole Results section. As suggested by this reviewer, we first present the analysis across all 8 CNVs and only then focus particularly on 16p11.2 and 22q11.2 deletions. The complete set of edits can be found in the improved manuscript draft with tracked changes.

Results:

We aimed to characterize how deletions **and duplications** of genomic sequences ~~at the 22q11.2 and 16p11.2~~ **across four genomic** loci impact brain asymmetry. For that purpose, we designed an analytic approach yielding ~~two~~ **eight** estimated linear discriminant analysis (LDA) models, each dedicated to a single CNV. In so doing, we isolated multivariate brain asymmetry patterns that discriminate between respective CNV carriers and controls (Table 1).

After exploring asymmetry patterns associated with **a broader portfolio of eight CNVs** ~~the two CNVs (deletions at 16p11.2 and 22q11.2 loci)~~, we **directed our analysis spotlight on two CNVs: deletions at 16p11.2 and 22q11.2 loci as the two most studied CNVs with high penetrance to neuropsychiatric disorders (Marshall et al., 2017; Sanders et al., 2015)**, ~~investigated whether similar patterns could be observed for a broader portfolio of eight CNVs building on prior research^{40,45}~~.

- Expanding the genetic correlation analysis to include rg's with ASD and SCZ could be helpful to either confirm previously found null rg, or to understand how the PT asymmetry genetically relates to these phenotypes both at the rare CNV level, as well as the level of common genetic variation.

We thank the reviewer for bringing up these interesting points. As suggested by the reviewer, we downloaded GWAS summary statistics from the Psychiatric Genomics Consortium (PGC) for both schizophrenia (SCZ) and autism spectrum disorder (ASD). Both resources represent the currently largest GWAS analysis on the respective disorder. We then quantified the genetic correlation between both disorders with left and right planum temporale volume as well as with planum temporale asymmetry. Genetic correlations with ASD ($r_{asym}=0.12$, $r_{left}=0.11$, $r_{right}=0.05$) were generally stronger than with SCZ ($r_{asym}=0.003$, $r_{left}=-0.03$, $r_{right}=-0.03$). However, none of the correlations reached significance (minimal p -value=0.31 for the genetic correlation between ASD and planum temporale asymmetry). We include these findings in the Results section.

Results:

Therefore, the genetic architecture of planum temporale asymmetry bore a degree of similarity to the GWAS signals pertaining to the left planum temporale volume but not those of the right planum temporale volume. **To complement these findings, we also quantified the genetic correlation with ASD and SCZ (cf. Methods). Genetic correlations with ASD (asymmetry: $R_g = 0.12$, left volume: $R_g = 0.11$, right volume: $R_g = 0.05$) were generally stronger than with SCZ (asymmetry: $R_g = 0.003$, left volume: $R_g = -0.03$, right volume: $R_g = -0.03$). However, none of the correlations reached significance (minimal p -value = 0.31 for the genetic correlation between ASD and planum temporale asymmetry).**

Discussion:

~~using the GWAS summary statistics~~ Similarly to our results, the authors did not find significant genetic correlations of planum temporale asymmetry with autism spectrum disorder, attention deficit hyperactivity disorder, schizophrenia, educational attainment, or intelligence ~~using the GWAS summary statistics~~.

Methods:

As the next step, we estimated genetic correlations between the three measurements of planum temporale, ASD, and SCZ. GWAS summary statistics for ASD and SCZ were obtained from the Psychiatric Genomics Consortium (PGC). Both resources represent the currently largest GWAS analysis on the respective disorder (Grove et al., 2019; Trubetskov et al., 2022). Calculating the genetic correlations was done using linkage disequilibrium (LD) score regression (LDSC, v.1.0.0)¹⁵¹, which is based on GWAS summary statistics generated in this study and does not require individual-level data.

Data resources:

Grove, J., Ripke, S., Als, T.D. et al. Identification of common genetic risk variants for autism spectrum disorder. *Nat Genet* 51, 431–444 (2019).

Trubetskov, V., Pardiñas, A.F., Qi, T. et al. Mapping genomic loci implicates genes and synaptic biology in schizophrenia. *Nature* 604, 502–508 (2022).

Figure 5

Single common variant associated with planum temporale asymmetry in genome-wide scanning

g) Genetic correlation between genetic basis (GWAS summary statistics) underlying left and right planum temporale volume, ~~as well as~~ planum temporale asymmetry, ~~as well as~~ ASD and SCZ. The left and right planum temporale volumes were significantly, yet imperfectly correlated. The remaining unexplained residual variance suggests a partly diverging genetic architecture. The genetic architecture underlying planum temporale asymmetry was significantly correlated only with the volume in the left hemisphere but not that of the right one. ~~We did not find a significant correlation with any of the disorders.~~

- I think the discussion section needs some improvement. Part of it covers a wide range of information about the planum temporale-associated phenotypes, without putting this in the context of the current study (and its starting point from the CNV analysis). I found some of its parts a bit farfetched (e.g. the role of music in society) for the current study.

We thank the reviewer for the suggestions. Based on these pointers, we have edited the Discussion in several places to increase fluency and sharpen the line of argument.

Discussion:

Clinical evidence indicates that lesions involving the left superior temporal gyrus, which extends into the planum temporale, give rise to Wernicke's aphasia. This tissue impairment typically leads to fluent but disordered speech production, impaired understanding of others' speech, and impaired silent reading^{60,61}. Further lesion and brain-imaging functional studies confirmed that the planum temporale is implicated in several crucial aspects of human communication, both spoken and gestural forms, that rely on a number of component neural processes, such as auditory and phonological processing, language comprehension⁶² and subvocal articulation^{63,64}. ~~A larger hemispheric divergence of the planum temporale surface was found among dyslexic children¹⁰. In addition, planum temporale surface asymmetry was also related to a family history of dyslexia⁸². Finally, reduced activity related to processing speech sounds in the planum temporale has been observed in children with developmental dyslexia⁸³. These human language-serving capacities contribute to our most evolved cognitive features in general⁶⁵ and are likely to be among the most strongly lateralized brain functions in the human brain².~~ Despite the necessity of the planum temporale for realizing language- and speech-related capacities, ~~in the intact central nervous system, its role extends well beyond language functions. As a current interpretational trend, the~~ planum temporale is currently viewed as a general ~~may be~~ advanced auditory signal processing hub responsible for representing and binding the location of sounds in space⁶⁶ as well as for pattern-matching operations between configurations of incoming sounds and previously stored auditory signals⁶³. Therefore, planum temporale circuits help solve a general neural computation that operates over many classes of stimulus types, including simple sound patterns⁶⁷, spectro-temporal modulations⁶⁸, speech, speech-like sounds, and speech-related cues such as prosody⁶⁹⁻⁷¹, but, notably, also tonal perception in music⁷²⁻⁷⁴. Indeed, a prior MRI study found increased cortical thickness and grey matter density of planum temporale in musicians⁷⁵. Consistently, our functional decoding analysis associated patterns centered on planum temporale asymmetry with functional keywords such as music, pitch, and tones. Therefore, our results suggest that CNVs that impact planum temporale asymmetry have downstream consequences on several lateralized functions highly evolved in humans, such as language and music perception.

In addition to having the probably most drastic functional asymmetry between the left and right brain, we are the only species that has given rise to a culture fueled by communication via articulated language⁶⁵. In our communication-centred society, language facilitates cultural inventions and has enabled many of today's societal institutions. ~~For example, language communication is fundamental to passing down knowledge resources from one generation to the next in schools and universities.~~ An integral mode of exchange between humans is the "universal language" of music. It is a cornerstone on which our feeling of togetherness and community rests. That is why, at the center of many, if not all, cultures studied so far is music as a fundamental part of social customs and rites⁷⁶. Music is thought to be the 'social glue' that enhances cooperation by strengthening feelings of unity and social belonging^{77,78}. Besides its cohesive benefit for our society, music can facilitate and intensify bonds in large groups and communities – a possible evolutionary extension of grooming in monkey societies⁷⁹. ~~Music-induced emotions thus transcend cultural boundaries⁷⁶ and involve ancient reward circuitry in the brain⁸⁰. From a neurochemical perspective, music can lead to dopamine release in the striatal system⁸¹. Therefore, music-induced emotional states can be effectively used in rituals and other religious, cultural, or social events to manipulate hedonic states⁸¹.~~ These elements illustrate why language and music have probably played important roles in the adaptation that the human brain and its genetic control underwent in evolutionary times. ~~Notably, studies have shown that CNVs may affect music perception (Ukkola-Vuoti et al., 2013).~~ In summary, it may be no accident that some of the most asymmetrical parts of the brain are also related to some of the most human-defining cognitive capacities in general. In this scenario, higher-order auditory functions and music-related practices are two prominent examples.

~~Based on the importance of the planum temporale for modes of human defining cognition, it is not surprising that alterations to planum temporale asymmetry (quantitatively and qualitatively different effects on one hemisphere relative to the other side) have far-reaching consequences for certain higher-order processes. For example, in terms of language skills, a larger hemispheric divergence of the planum temporale surface was~~

~~found among dyslexic children¹⁰. In addition, planum temporale surface asymmetry was also related to a family history of dyslexia⁸². Finally, reduced activity for processing speech sounds in the planum temporale has been observed in children with developmental dyslexia⁸³.~~

minor comments

introduction

- page 4, last paragraph. What is the motivation to focus on 16p11.2 and 22q11.2 deletions to start with? either here or at the beginning of the results, it would be helpful if you guided the reader as to why these two CNVs have been analyzed separately in the first place.

We thank the reviewer for the comment. We extend the description of CNVs in general as well as 16p11.2 and 22q11.2 in particular. These two CNVs carry the highest risk for schizophrenia and autism spectrum disorder, respectively. At the same time, these two CNVs are well known for their impacts on language - ability associated with brain lateralization.

Introduction:

Although CNVs are well known to impact some of the most lateralized cognitive functions, including language capacity⁴¹, how CNVs affect structural brain asymmetry has not yet been explored. 16p11.2 and 22q11.2 deletions stand out as the two “relatively” frequent CNVs with large effects on risk for neuropsychiatric disorders⁴⁵. The most substantial increase in risk for schizophrenia (SZ) is associated with 22q11.2 deletions (30 to 40-fold increase) (Marshall et al., 2017), while 16p11.2 deletions confer notably elevated risk (10-fold increase) for autism spectrum disorder (ASD) (Sanders et al., 2015). In terms of language impairments. ~~To illustrate the apparent CNV-induced language impairments~~, 77% of children and 50% of adults carrying a 16p11.2 deletion meet the criteria for childhood apraxia of speech⁴². Furthermore, 95% of children with a 22q11.2 deletion are diagnosed with speech-language disorders⁴³. Speech and language delays are hallmark features of CNVs in pediatric clinics; perhaps the earliest symptoms for which children with a CNV get clinically referred in the first place⁴⁴.

Results:

After exploring asymmetry patterns associated with a broader portfolio of eight CNVs ~~the two CNVs (deletions at 16p11.2 and 22q11.2 loci)~~, we directed our analysis spotlight on two CNVs: deletions at 16p11.2 and 22q11.2 loci as the two most studied CNVs with high penetrance to neuropsychiatric disorders⁴⁵, ~~investigated whether similar patterns could be observed for a broader portfolio of eight CNVs building on prior research^{40,45}~~

results

- page 5 "Brain asymmetry effects induced by 16p11.2 and 22q11.2 deletions": to which extent are the LDA models good fit? i.e. sensitivity and specificity are not reported.

We thank the reviewer for bringing up this important point. We evaluated all LDA models using the ROC curve (receiver operating characteristic curve). The ROC curve is a plot of sensitivity on the y-axis against (1-specificity) on the x-axis for varying values of the threshold for group distinction. The AUC (Area under the ROC Curve) metric can serve as a single-number summary of model fit. Across the eight analyzed CNVs, the maximal AUC was 0.93 (1q21.1dup), while the minimal AUC was 0.71 (15q11.2dup). We now comment on the goodness of fit in the Results. The ROC analysis is now part of the Supplementary Material.

Results:

For that purpose, we designed an analytic approach yielding eight estimated linear discriminant analysis (LDA) models, each dedicated to a single CNV. In so doing, we isolated multivariate brain asymmetry patterns that discriminate between respective CNV carriers and controls (Table 1). All eight LDA models were effective at performing the CNV-control discrimination task as suggested by ROC analysis (Sup. Fig. 1), where the maximal under the ROC curve was 0.93 (1q21.1dup), while the minimal area was 0.71 (15q11.2dup).

Supplementary Figure 1

All eight CNVs can well separate CNV from control participants based on brain-imaging features.

We leverage the ROC curve (receiver operating characteristic curve) analysis in order to evaluate the discrimination performance of eight LDA models. Each LDA model was aimed at classifying between dedicated CNVs and controls. The depicted ROC curve is a plot of the True Positive Rate (sensitivity) on the y-axis against the False Positive Rate (1-specificity) on the x-axis for varying values of the threshold. The AUC (Area under the ROC Curve) metric can serve as a single-number summary of model fit. Across the eight analyzed CNVs, the maximal AUC was 0.93 (1q21.1dup), while the minimal AUC was 0.71 (15q11.2dup). All AUC values are well above the chance level (AUC=0.5), suggesting a good model fit for all eight LDA models.

- pages 5-6 "Direction of magnitude of induced asymmetry indices": for the planum temporale asymmetry, 16p11.2 and 22q11.2 deletion carriers showed an increased leftward asymmetry compared to controls. This is somehow counterintuitive, as the leftward lateralization in the general population is usually interpreted to the leftward asymmetry of language function? Given the language deficit phenotypes these deletion carriers present, I would have expected an opposite pattern (i.e. less lateralized PT? or rightwards?). Nonetheless, structural PT asymmetry cannot be interpreted as an index of individual variability in language dominance (e.g. see Tzourio-Mazoyer et al. 2018, DOI: 10.1007/s00429-017-1551-7). May be worth discussing further in the discussion.

We thank the reviewer for the stimulating comment. From Figure 1c, it is apparent that CNVs can, on average, both decrease or increase the asymmetry of planum temporale (PT). This interesting observation suggests that any change in PT asymmetry might be associated with language performance. Furthermore, the observed increased PT asymmetry in 16p11.2 and 22q11.2 deletions stems from, on average, increased PT volumes in carriers of these two CNVs compared to controls (Fig. 3b). This volume increase is different between the left and right hemispheres, resulting in abnormal asymmetry. As pointed out, these asymmetries are not a marker of inter-individual variability in language dominance. We now comment on these observations in the Discussion.

Discussion:

Hence, in neurotypical individuals, the planum temporale is highly structurally lateralized, with 65% of all individuals showing larger left planum temporale; and only 11% of individuals showing larger right planum temporale volume^{47,59}. Interestingly, we found that CNVs can lead to both increased and decreased planum temporale asymmetry, suggesting that different deviations might be associated with language performance.

However, it is important to note that planum temporale asymmetry is not a marker of inter-individual variability in language dominance (Tzourio-Mazoyer et al., 2018).

- Related, from figure 2b and Cohen's d's reported in page 6 (first paragraph) it seems that CNV carriers have larger left *and is not a marker of inter-individual variability in language dominance* right PTs, maybe reflecting a more global effect. I.e. larger brains tend to have larger asymmetries? (see Williams et al. 2022 for a discussion, DOI: 10.1016/j.neuroimage.2022.119118). You have already performed a sensitivity analysis to assess the effect of deconfounding additional factors, and shown that the loadings are highly correlated (Supplementary Figure 3), it may however help understand where the effects are coming from (i.e. maybe they affect differently left and right?).

We thank the reviewer for bringing up this interesting remark. As pointed out, we performed several sensitivity analyses, including those focused on the effect of controlling for total brain volume. As a result, none of the brain-imaging-derived results presented in this set of analyses or findings is influenced by the effects of CNVs on total brain volume. Indeed, we have previously demonstrated that the regional and global effects of CNVs on neuroanatomy are independent of one another (Martin-Brevet et al., 2018). As noted above, the observed increased PT asymmetry in 16p11.2 and 22q11.2 deletions stems from, on average, increased PT volumes in carriers of these two CNVs compared to controls (Fig. 3b). This volume increase is different between the left and right hemispheres, resulting in abnormal asymmetry and suggesting different effects of CNVs on each hemisphere.

- page 6, Figure 3a-c: given that the LDA coefficients for 22q11.2 del are a subset of the 16p11.2, I don't find it surprising that there is a high overlap in the functionally associated terms for these regions.

We thank the reviewer for highlighting this lack of clarity on our side. The functional association analysis was based on all LDA coefficients (i.e., 65 LDA coefficients for 65 brain regions) for both CNVs. Although 3 of these 65 LDA coefficients are of similar magnitude and direction, it might not be the case for the remaining 62 LDA coefficients. Therefore, due to the similarity of the strongest coefficients, similar functional associations might be expected but are not warranted. We now clarify this potential confusion in the Results section.

Results:

We subsequently turned to NeuroSynth⁴⁸ to mine candidate functional explanations for the distilled brain asymmetry patterns that separated CNV carriers from controls. To this end, we mapped the **full collection of 65 obtained** (absolute) LDA coefficients onto both hemispheres, with the same coefficient for homologous regions. In other words, we created a whole-brain map indicative of CNV carriership based on anatomically localized classifier coefficients.

- page 7 "Comparison of asymmetry patterns across eight CNVs" it would be helpful to know how well these LDA models are.

This point is answered above in the first answer of the ## results section.

- page 7, when presenting the role of PT and fusiform asymmetries across the 8 CNVs, it is now misleading that they contributed 3/8 CNVs. It would be more fair to say that they contributed to one additional CNV apart from the deletions of 16p and 22q (which you have already presented in the sections above).

We thank the reviewer for this remark. This potential confusion is now solved due to the reorganization of the Results section. Since we now first present results across 8 CNVs, the statement that PT contributed to 3/8 CNVs is valid.

- Figure 4b, suggestion: may be worth having a boxplot rather than barplot to show the spread of the LDA coefficients across the 8 CNVs. Maybe also labelling the points that had significant loadings.

We thank the reviewer for the valuable suggestion. We updated the figure as requested. The boxplot now depicts LDA coefficients for all eight CNVs. At the same time, the marker type highlights significant LDA coefficients.

Old figure:

New figure:

Figure 1

Eight key CNVs lead to unique effects on brain asymmetry that spotlight the planum temporale

b) Overall strongest coefficients across eight LDA models.

The boxplots depict The Y-axis depicts mean LDA coefficients across the 8 CNVs. Planum temporale shows the strongest mean magnitude, followed by lobule VIIIb of the cerebellum and putamen. Lighter symbols represent deletions, while darker symbols represent duplications. Star denotes a significant coefficient based on the bootstrap significance test. Error bars correspond to a 95% confidence interval of the coefficient mean.

- page 7, lines 256-257: "CNVs known to have deleterious consequences on language performance"... how is 15q11.2 related to language?

We thank the reviewer for this question. Symptoms associated with 15q11.2 BP1-BP2 CNV have a much smaller effect size on neurodevelopmental symptoms than the 3 other loci investigated in our study. Meta-analysis across multiple medical and psychiatric diagnoses (Jønych et al., 2019) shows that the 15q11.2 deletion has a significant but mild effect on risk for neurodevelopmental disorders, schizophrenia, epilepsy and congenital heart disease. Data from population studies further indicate that deletion carriers unaffected by severe psychiatric and neurodevelopmental disorders have an increased prevalence of dyslexia and dyscalculia (Kendall et al., 2017). The reciprocal genomic duplications in this region have not yet formally been associated with developmental or psychiatric symptoms (Jønych et al., 2019). In summary, although the effect sizes associated with deletions/duplications of 15q11.2 regions are smaller compared to those associated with 1q21.1, 16p11.2, or 22q11.2, speech and language might be impaired in the respective carriers.

Literature:

- Jønych, A. E. et al. Estimating the effect size of the 15Q11.2 BP1–BP2 deletion and its contribution to neurodevelopmental symptoms: recommendations for practice. *J Med Genet* 56, 701–710 (2019).
- Kendall KM, et al. Cognitive performance among carriers of pathogenic copy number variants: analysis of 152,000 UK Biobank subjects. *Biol Psychiatry*. 2017;82(2):103-110.

- page 8, second paragraph: I'm not entirely sure, but would it make sense to perform a clustering analysis based on the LDA weights / correlation in Figure 4d? If possible, this could inform about how the different CNVs cluster with regard to brain asymmetry, and whether there is a unique or multiple clusters.

We thank the reviewer for the suggestion. We have now added a hierarchical clustering of the obtained LDA coefficients using the Ward variance minimization algorithm. The resulting dendrogram is now part of Figure 1d (previously 4d). Results highlight three main clusters: 1q21.1 and 16p11.2 duplications, all deletions, 15q11.2 and 22q11.2 duplications. Such a separation provides further evidence of the opposing effects of deletions and duplications. We now comment on these findings in the Results section.

Results:

Similar to prior analyses of regional volumes, we observed mirroring effects on regional asymmetry conferred by the deletion and duplications at the same loci. The strongest mirroring effect emerged between deletion and duplication at the 16p11.2 locus ($R = -0.45$, $P_{spin} = 0.01$), followed by the 1q21.1 locus ($R = -0.39$, $P_{spin} = 0.02$). To summarize these comparisons, we submitted the model coefficients to hierarchical clustering with the Ward similarity measure. This post-hoc analysis revealed three clusters: 1q21.1 and 16p11.2 duplication, our four deletions, 15q11.2 and 22q11.2 duplication. Such a separation provides further evidence of the opposing effects of deletions and duplications.

Figure 4

Eight key CNVs lead to unique effects on brain asymmetry that spotlight the planum temporale

d) Comparison of brain-wide asymmetry patterns. We calculated Pearson's correlation to quantify the similarity between every pair of LDA coefficient sets from the eight models. Asterisk denotes FDR-corrected P-values obtained from spin permutation. Hierarchical clustering of model-specific coefficients obtained per CNV display distances (similarities) among studied asymmetry patterns based on Ward's method. Distinct clusters separating deletions and duplications provide further evidence of their opposing effects.

- page 8: gene names should be in italics.

All gene names are now in italics.

- page 8: the SNP associated with PT asymmetry (rs41298373) is coding non-synonymous SNP. It also has a particular LD structure around this hit, reflected by the lack of other SNPs in this locus associated with PT asymmetry (as shown in the Manhattan plot in Figure 5a)

We thank the reviewer for these remarks. We incorporated them into the manuscript.

Results:

Planum temporale asymmetry turned out to relate to a single locus on chromosome 10p14 (rs41298373, $P = 3.31 \times 10^{-17}$) (Fig. 5a). This single significant SNP is a coding non-synonymous variant with a particular LD structure reflected by the lack of other SNPs in this locus. The SNP significance was confirmed by several sensitivity analyses directed at the definition of the asymmetry index (Sup. Fig. 1).

-page 9, lines 311-316: it is not clear to me what do the left and right PT GWAS signal add with regard to the loci that are not significant in the AI GWAS. You mention (lines 323-324) that the CNV effect is driven by a larger left PT, but do not provide any interpretation as to how these signals could contribute to the observed effects.

We thank the reviewer for bringing up this potential confusion. We present findings from the left and right PT GWAS in order to showcase the different genetic architecture of left and right PT. These observations are

mentioned in the Discussion. We shortened this section to sharpen the line of argument. Moreover, based on the suggestions, we now highlight similar observations for SNP and CNV analyses. However, it is not possible to relate the effects of localized high-effect size CNVs with pleiotropic small-effect size SNPs.

Next, we explored if there are similar genome-wide association profiles (including the significant association with the *ITIH5* gene) for the raw volumes of the left and right planum temporale (Sup. Fig. 2). Notably, the planum temporale asymmetry-related *ITIH5* gene was only observed in SNPs associated with the left planum volume (rs41298373 $P = 5.64 \times 10^{-13}$), but not the right planum temporale. Based on GWAS of the left planum temporale volume, we found 726 significant candidate SNPs that mapped to eight genomic loci. After functional filtering for gene mapping, we obtained eight lead SNPs that are assumed to entail variations of ten protein-coding genes (*CDC42*, *FAM172A*, *ITIH5*, *BCL11B*, *LRCH1*, *ENPP2*, *SEMA3D*, *WNT4*, *TLN2*, *KIAA0825*) (Fig. 5f). ~~These ten protein-coding genes (*CDC42*, *FAM172A*, *ITIH5*, *BCL11B*, *LRCH1*, *ENPP2*, *SEMA3D*, *WNT4*, *TLN2*, *KIAA0825*) were distributed across eight chromosomes, with *WNT4* and *CDC42* lying on the first chromosome and *FAM172A* and *KIAA0825* on the fifth chromosome.~~

Incongruent with the GWAS signals pertaining to the left planum temporale, our GWAS of the right planum temporale volume yielded 368 candidate SNPs in four genomic loci. We further refined this SNP set to four lead SNPs related to six genes (*DMRTA2*, *FAF1*, *ATP1B3*, *TFDP2*, *C14orf177*, and *C14orf64*) – none of which was related to left planum temporale volume (Fig. 5f). ~~Two (*DMRTA2*, *FAF1*) of these six protein-coding genes lie on the first chromosome. *ATP1B3* and *TFDP2* are part of the third chromosome. Finally, *C14orf177* and *C14orf64* lie on chromosome fourteen.~~ Notably, we observed ~~more larger CNV and SNPs associated with effects on~~ the left planum temporale compared to its right hemispheric homologue. ~~Such an observation is consistent with larger CNV effects on left planum temporale.~~

- page9, third paragraph: the genetic correlation between left and right PT you report is exactly the same as the one in Carrion-Castillo et al. (2020, DOI: 10.1016/j.cortex.2019.11.006).

We thank the reviewer for spotting the missing citation, which we now provide in the Results section.

We found a notable yet incomplete genetic correlation between the genetic underpinnings that influence the left and right planum temporale volumes in our UK Biobank participant sample ($R_g = 0.85$, $P = 1.53 \times 10^{-128}$) (Fig. 5g) - ~~an identical value as prior research (Carrion-Castillo et al., 2020).~~

- The much larger UK biobank sample size that you have used in the current analysis (N ~30,000 vs N~18,000) could be used to explore whether PT AI (or left PT) are genetically correlated with ASD and SCZ (the previous study tested this and found no evidence, but your current GWAS would be better powered). If such a genetic correlation were found, it would be a nice way of linking back to the CNV phenotypes as well.

This point is answered above in the third answer of the ## major comments section.

discussion

- this section reads as a long part about the planum temporale followed by some context for the other identified asymmetries. I think some more context about the CNVs would be helpful to guide the reader

We thank the reviewer for the suggestions. The reduction of the planum temporale description is described above in the fourth answer of the ## major comments section. In addition, we provide more context about CNVs.

Discussion:

~~Our functional annotations of CNV-induced brain asymmetry patterns revealed links to speech perception, language comprehension, and hearing capacities.~~ Among the distilled morphological patterns, the planum temporale asymmetry played a central role. More specifically, the planum temporale was the most

commonly and most strongly affected region in our assessment of eight CNVs. Prior studies identified volumes of left and right planum temporale as sensitive to gene dosage effects and as a source of morphological variation across CNV (Martin-Brevet et al., 2018; Modenato et al., 2021). Planum temporale is a triangular-shaped region which occupies the superior temporal plane posterior to Heschl's gyrus¹⁰ and is part of a neural circuit that passes through left-hemisphere language regions³. This region forms a part of the perisylvian area that is known to show the greatest structural left-right differences in the human brain in general¹⁶. Hence, in neurotypical individuals, the planum temporale is highly structurally lateralized, with 65% of all individuals showing larger left planum temporale; and only 11% of individuals showing larger right planum temporale volume^{47,59}. ~~From a broader perspective, this advanced processing region is part of a neural circuit that passes through left-hemisphere language regions³.~~ In terms of functional lateralization, there is a possibility that microstructural and not macrostructural features are more important (Ocklenburg et al., 2018). Postmortem studies highlighted the role of intrinsic microcircuitry organization (Galuske et al., 2000). However, the role of micro- and macro-structure is beyond the scope of this article since the used structural MRI signal captures contributions from various structural components within a voxel and is not able to disentangle microstructure properties (Song et al., 2022).

Discussion:

The role of language in developmental periods is fundamental, from prenatal development⁸⁴ to adulthood. Thus, language impairments will have lifelong consequences for the ability to meet the demands and challenges of contemporary societies. As an example, language impairment may contribute to reported lower educational attainment and decreased ability to earn income in adult life in CNV carriers (Kendall et al., 2019). Furthermore, ~~As expected,~~ language-related symptoms are present in many neurodevelopmental disorders. CNVs at our four loci present some of the most frequent risk factors for these disorders (up to a 40-fold increase in the risk (Marshall et al., 2017)). A line of evidence shows that the sources of these specificities co-localize to the planum temporale. A reduction of planum temporale grey matter volume asymmetry has been reported in people with autism spectrum disorder (ASD)^{85,86}. A reduction of the left planum temporale has been proposed to relate to delays in language acquisition, which are commonly observed clinical features in ASD patients⁸⁷. Consistent with these findings, a study in newborns reported a strong association between the growth rate in the left planum temporale over the first 3 weeks of life and language scores at 12 months of age⁸⁸.

Discussion:

The average leftward asymmetry of the planum temporale is established even before birth. It can be observed starting from the 31st week of gestation, as evidenced by post-mortem studies of fetuses⁹⁵. Furthermore, perisylvian regions, including the planum temporale, were the only regions to be asymmetric based on in-vivo MRI of preterm newborns from 26 to 36 weeks of gestational age⁹⁶. Given this repeated observation, average leftward asymmetry of the planum temporal appears to be encoded in the human genome, yet by currently unknown mechanisms. ~~Although detailed knowledge of brain structure development in CNV carriers over the lifespan is lacking,~~ macrocephaly was observed already in utero in a 1q21.1 distal duplication carrier (Verhagen et al., 2015). A recent study pointed to a significant heritability of planum temporale structural asymmetry of around 14%⁵³.

Discussion:

. As evidenced by the strong effects on cerebellum lobule VIIIb, the CNV effects are not limited to the cerebrum. A prior study found several cerebellar regions (vermis lobule VIII-X and cerebellar cortex) to be highly sensitive to CNVs - potentially due to the cerebellum's protracted development (Sathyanesan et al., 2019). Notably, individuals with the most functionally asymmetric cerebrums also possess the most asymmetric cerebellums in general⁹⁷.

Discussion:

Finally, our whole-brain asymmetry analysis also spotlighted the temporal fusiform cortex to be associated with 16p11.2 and 22q11.2 deletions. Fusiform gyrus was reported to be among regions markedly altered across CNVs (Modenato et al., 2021). ~~left and right fusiform gyri display functional asymmetries~~ From a

functional perspective, with asymmetrical contribution, the right fusiform gyrus is known to subserve conscious processing of faces, and the left homologue engages in more general visual perception and object recognition^{110,111}.

- page 10, lines 357-358: "(...) which demonstrated that the genetic basis of left and right planum temporale volume is partly diverging". This is not new, as it was already explored in detail by Carrion-Castillo et al. (2020, DOI: 10.1016/j.cortex.2019.11.006). Please rephrase, e.g. replace "demonstrated" with "confirmed" or similar.

We reformulated the sentence as suggested.

Discussion:

This converging result motivated a GWAS investigation of the planum temporale volume, which confirmed demonstrated that the genetic basis of left and right planum temporale volume is partly diverging.

- page 11, first paragraph: I don't see that you need all this information about music, its role as "social glue" etc. It may be relevant, given the phenotypes of CNV carriers, but no information in this respect is provided.

We thank the reviewer for highlighting this point. In line with the recent re-evaluation of planum temporale functionality, we argue that alterations of planum temporale asymmetry are not relevant only to language abilities but to all auditory processing functions. Furthermore, the revealed CNV-induced brain asymmetry patterns were linked with functional keywords, such as sounds, acoustic, tones or pitch, which calls for carving out the link between language, music, brain, and society. Therefore, the paragraph serves a broader purpose in addition to providing information about music and its role as 'social glue'. It discusses the unique characteristics of the human brain, its functional asymmetry, and the role of both language and music in shaping human culture and society. This broader context helps set the stage for understanding the potential impact of CNVs on more complex aspects of human cognition and behavior. By providing insight into the significance of language and music, the paragraph aims to highlight the multifaceted nature of human cognitive capacities. Given the importance, it is not surprising that these capacities are under genetic control, which has a direct link to CNVs as tools of genetic alteration. In line with the suggestions, we have shortened the paragraph in several places to increase fluency and sharpen the line of argument while also adding links to CNVs.

Discussion:

In addition to having the probably most drastic functional asymmetry between the left and right brain, we are the only species that has given rise to a culture fueled by communication via articulated language⁶⁵. In our communication-centred society, language facilitates cultural inventions and has enabled many of today's societal institutions. ~~For example, language communication is fundamental to passing down knowledge resources from one generation to the next in schools and universities.~~ An integral mode of exchange between humans is the "universal language" of music. It is a cornerstone on which our feeling of togetherness and community rests. That is why, at the center of many, if not all, cultures studied so far is music as a fundamental part of social customs and rites⁷⁶. Music is thought to be the 'social glue' that enhances cooperation by strengthening feelings of unity and social belonging^{77,78}. Besides its cohesive benefit for our society, music can facilitate and intensify bonds in large groups and communities – a possible evolutionary extension of grooming in monkey societies⁷⁹. ~~Music induced emotions thus transcend cultural boundaries⁷⁶ and involve ancient reward circuitry in the brain⁸⁰. From a neurochemical perspective, music can lead to dopamine release in the striatal system⁸¹. Therefore, music-induced emotional states can be effectively used in rituals and other religious, cultural, or social events to manipulate hedonic states⁸¹.~~ These elements illustrate why language and music have probably played important roles in the adaptation that the human brain and its genetic control underwent in evolutionary times. Notably, prior studies have shown that CNVs may affect music perception (Ukkola-Vuoti et al., 2013). In summary, it may be no accident that some of the most asymmetrical parts of the brain are also

related to some of the most human-defining cognitive capacities in general. In this scenario, higher-order auditory functions and music-related practices are two prominent examples.

- page 12, lines 458-460: "Notably..." the association of rs41298373 with left PT but not right PT was already reported by Carrion-Castillo et al. (2020).

We modified our statement to highlight the fact that these associations were already reported.

Notably, our whole-genome results confirm previous findings further sharpen the picture that this locus is significantly associated only with the left, but not the right, planum temporale volume⁵³.

methods

- page 16, line 632: why are there 290 controls included as opposed to 1296 controls included in Modenato et al. 2021? please specify how the subset for the current study has been defined.

We thank the reviewer for highlighting the lack of clarity. We used 290 controls from the clinical ascertainment cohort in Modenato et al. 2021. We make it clear in the Method section. As this number is significantly higher than any group of CNV carriers, the inclusion of more controls did not lead to different results.

In total, our dataset consisted of volumetric asymmetries derived from structural MRI brain scans of 842 subjects in total: 552 CNV carriers and 290 controls from the clinical ascertainment not carrying any CNV (Table 1).

- page 16, paragraph 3 (starting line 633): I found this paragraph useful, and would have appreciated it in the introduction to justify why these CNVs are of interest.

We thank the reviewer for the valuable suggestion. We moved the paragraphs into the Introduction section. The updated introduction is available above in the first answer of the ## major comments section.

- page 18: were the LDA models somehow evaluated?

This point is answered above in the first answer of the ## results section.

Table 1

- page 24: Table 1 caption. When defining the ICD10 codes that correspond to each diagnosis labels are flipped: line 927 should read "SCZ" instead of "ASD", line 929 should read "ASD" instead of "SCZ".

The typos were corrected.

~~SCZASD~~ diagnosis in the UK Biobank corresponded to the ICD10 code, including schizophrenia, schizotypal and delusional disorders (F20-F29). ~~ASDSEZ~~ diagnosis in the UK Biobank corresponded to the ICD10 code that included diagnoses of childhood autism (F84.0), atypical autism (F84.1), Asperger's syndrome (F84.5), other pervasive developmental disorders (F84.8), and pervasive developmental disorder, unspecified (F84.9).

- page 24: Table 1 caption. Why are the UKB diagnosis definitions included here? If part of the dataset is from the UKB, please specify how many. For the non-UKB samples, do they also have diagnosis information? Also, given the age differences between CNV carriers and controls, specify age range in addition to sd?

We thank the reviewer for highlighting the potential lack of clarity. As described in the Methods section: "The participant sample used in this study represents a combination of a carefully collected multi-site clinical cohort and the UK Biobank. Specifically, we grouped 295 CNV carriers identified in the UK Biobank and 257 CNV

carriers from the clinical cohorts.” Therefore, the information for non-UKB samples comes from the respective hospitals, while the diagnosis for UKB samples is from respective ICD 10 codes. We make this distinction clear in the description.

CNV loci chromosome coordinates are provided with the number of genes encompassed in each CNV and a well-known gene for each locus to help recognize the CNV. **ASD and SCZ diagnoses for clinically ascertained CNV carriers were obtained from respective data acquisition sites.** ASD diagnosis in the UK Biobank corresponded to the ICD10 code, including schizophrenia, schizotypal and delusional disorders (F20-F29). SCZ diagnosis in the UK Biobank corresponded to the ICD10 code that included diagnoses of childhood autism (F84.0), atypical autism (F84.1), Asperger’s syndrome (F84.5), other pervasive developmental disorders (F84.8), and pervasive developmental disorder, unspecified (F84.9). Abbreviations, Del: deletion; Dup: duplication; ASD: autism spectrum disorder; SZ: schizophrenia; chr: chromosome; Age: mean age; SD: standard deviation; nGenes: number of genes.

We added the age range as suggested.

Loci	Chr (hg19) start-stop	nGenes (Gene)	Type	Subjects	Age (SD) [range]	Sex (M/ F)	ASD SZ
1q21.1	chr1	7	Del	32	40 (17)[9-73]	14 / 18	0 0
	146.53-147.39	CHDIL	Dup	27	44 (15)[8-66]	11 / 16	3 0
15q11.2	chr15	4	Del	110	55 (7)[40-69]	50 / 60	0 0
	22.81-23.09	CYFIP1	Dup	144	54 (7)[40-69]	67 / 77	0 0
16p11.2	chr16	27	Del	82	19 (15)[7-63]	48 / 34	10 0
	29.65-30.20	KCTD13	Dup	69	32 (15)[8-63]	39 / 30	7 1
22q11.2	chr22	49	Del	66	19 (13)[6-66]	32 / 34	8 2
	19.04-21.47	AIFM3	Dup	22	26 (19)[8-66]	12 / 10	2 0
Controls				290	26 (14)[6-64]	162 / 128	1 0

Reviewer #2 (Remarks to the Author):

Review of Nature Communications Manuscript NCOMMS-23-22437-T

In their manuscript entitled “Using rare genetic mutations to revisit structural brain asymmetry” Kopal and co-workers used UK Biobank data and other data to investigate the role of rare genetic mutations in structural hemispheric asymmetries in the human brain. The most noteworthy result is that asymmetry of the planum temporale, a well-known brain area relevant for language processing was especially susceptible to deletions and duplications of specific gene sets. I think that the work could be of significance to neuroscience and related fields. The established literature on hemispheric asymmetries mostly focused on common variants, with few exceptions. Moreover, the established literature focused mostly on functional hemispheric asymmetries, not structural hemispheric asymmetries, as in this work. The work supports the conclusions and claims, and I do see any flaws in the data analysis, interpretation, and conclusions that would prohibit publication. The methodology is sound, and the work meets the expected standards in the field.

In general, I could see it published in NCOMMS, but have a few comments that the authors may wish to address in a revised version of the manuscript.

For now, I recommend:

Minor revisions

We are thankful for the enthusiastic feedback.

Specific points:

General:

Please check whether the font size is uniform throughout the manuscript. There were some somewhat irritating changes in font size in the manuscript (e.g., between the first and the second sentence in the abstract).

We thank the reviewer for highlighting the inconsistency in our Abstract. The font size is now uniform across the whole manuscript.

Introduction

Statement “Enabled by a multicohort machine learning approach, we quantitatively dissected the impact on brain asymmetry of eight CNVs: deletions and duplications at 1q21.1 distal, 15q11.2 BP1-BP2, 16p11.2 proximal, and 22q11.2 proximal loci, with a particular focus on deletions at 16p11.2 and 22q11.2 loci as the two most studied CNVs with high penetrance to neuropsychiatric disorders.”

The readers may not be immediately familiar with these specific CNVs. It would be great if the authors could provide more information on them such as likely associated genes and likely associated functions as well as references to the published literature that may relate them to asymmetries. In general, the authors should explain a bit more why these specific CNVs were chosen for studying brain asymmetries.

We thank the reviewer for highlighting this important point. We extended the description of used CNVs in the Introduction. In addition, we also included more details in the Methods section.

Intro:

CNVs at the 22q11.2, 16p11.2, 1q21.1, and 15q11.2 genomic loci are among the most commonly identified risk factors for neuropsychiatric disorders in pediatric clinics (Moreno-De-Luca 2013; Crawford et al., 2019). These deletions and duplications of genomic sequences strike a balance between occurrence in the population and their strong biological consequences. In other words, while these CNVs are rare, especially compared to single nucleotide polymorphisms studied in GWAS, these genetic alterations are frequent enough (between 1 in 500 and 1 in 4000 in the general population) so that we can begin to carry out across-CNV investigations in population datasets.

Specifically, these CNVs are now being understood to exert body-wide implications, including the cardiovascular, endocrine, skeletal and nervous systems (Auwerx et al., 2022; Kopal et al., 2023), with

deteriorating consequences to everyday life (Kendall et al., 2021). The substantial downstream consequences suggest that CNVs may serve as a sharp imaging-genetics tool for interrogating the effects of genetic alterations on brain physicality and behavioral differentiation³⁵. ~~Studies have already started to delineate the effects of CNVs on brain structure^{36,37} and function³⁸⁻⁴⁰.~~ Although CNVs are well known to impact some of the most lateralized cognitive functions, including language skills⁴¹, how CNVs affect structural brain asymmetry has not yet been explored. 16p11.2 and 22q11.2 deletions stand out as the two “relatively” frequent CNVs with large effects on risk for neuropsychiatric disorders⁴⁵. The most substantial increase in risk for schizophrenia (SZ) is associated with 22q11.2 deletions (30 to 40-fold increase) (Marshall et al., 2017), while 16p11.2 deletions confer notably elevated risk (10-fold increase) for autism spectrum disorder (ASD) (Sanders et al., 2015). In terms of language impairments, ~~To illustrate the apparent CNV-induced language impairments,~~ 77% of children and 50% of adults carrying a 16p11.2 deletion meet the criteria for childhood apraxia of speech⁴². Furthermore, 95% of children with a 22q11.2 deletion are diagnosed with speech-language disorders⁴³. Speech and language delays are hallmark features of CNVs in pediatric clinics; perhaps the earliest symptoms for which children with a CNV get clinically referred in the first place⁴⁴.

Methods:

The here-examined CNVs are among the most commonly studied CNVs⁴⁵. ~~Deletions and duplications of 1q21.1, 15q11.2, 16p11.2, and 22q11.2 represent rare but reoccurring risk factors for neuropsychiatric disorders identified in pediatric clinics^{114,115}. That is why~~Specifically, these CNV loci were ~~also~~ selected in many other studies, such as an independent research study conducted by The Enhancing NeuroImaging Genetics through Meta-Analysis copy number variant (ENIGMA-CNV) on cognitive, psychiatric, and behavioral manifestations³⁷. The 22q11.2, 16p11.2, 1q21.1, and 15q11.2 genomic loci encompass 49, 27, 7 and 4 genes, respectively (Mefford et al., 2008; Golzio et al., 2012; Jonas et al., 2014). ~~This set of deletions and duplications of genomic sequences strikes a balance between occurrence in the population and their effect size. In other words, while the selected CNVs are rare (especially compared to single nucleotide polymorphisms), they are frequent enough so that we can start studying large enough sample sizes that allow for across-CNV comparison in the first place. At the same time, this class of CNVs has been shown to detrimentally affect cognition and raise the risk for psychiatric conditions³⁷.~~ Our CNV carriers in our clinical dataset did not carry any other known large CNV.

I am not sure about the journal’s policy on citing preprints that have not undergone peer review, but the following preprint focused on rare variants in handedness, a form of functional hemispheric asymmetries that may be related to structural hemispheric asymmetries may be worthwhile to include in the intro:
<https://www.biorxiv.org/content/10.1101/2023.05.31.543042v1>

Moreover, the following study also used whole exome sequencing in functional hemispheric asymmetries and may be relevant:

Kavaklioglu T, Ajmal M, Hameed A, Francks C. Whole exome sequencing for handedness in a large and highly consanguineous family. *Neuropsychologia*. 2016 Dec;93(Pt B):342-349.

We thank the reviewer for highlighting these important resources. We added a paragraph to the Results section discussing the relationship between handedness and CNV-induced changes to brain asymmetry.

Results

To supplement the functional profiling using NeuroSynth, we investigated if derived asymmetry patterns are associated with handedness. Although how genetic variants contribute to handedness remains a conundrum (Kavaklioglu et al., 2106), recent studies point to the role of rare variants in handedness (Schijven et al., 2023). Therefore, we probed for the association between the derived asymmetry pattern and handedness in the 36,000 UK Biobank participants but did not find any robust association (Sup. Fig. 2).

It would be great if the authors could present a hypothesis at the end of the introduction. While I understand that this is largely a data-driven study, there must have been something that pointed the authors into the general direction of conducting this study.

We thank the reviewer for the valuable suggestion. We included a hypothesis and the end of our Introduction section.

Introduction

In 552 CNV carriers and 290 non-carriers, we systematically explored CNV-specific brain asymmetry patterns using a machine learning toolkit, including linear discriminant latent factor modeling. Specifically, we isolated multivariate brain asymmetry patterns that distinguish between respective CNV carriers and controls based on the asymmetry in volumes captured by reference atlases. **Given the widely acknowledged associations of CNVs with neuropsychiatric and especially speech-language disorders (Solot et al., 2019; Moreno-De-Luca et al., 2013), we hypothesized that CNVs affect structural asymmetry in brain regions associated with lateralized functions, including language. In this way** Our data-led imaging-genetic **results study** sheds new light on why so many CNVs are clinically associated with performance decay in higher cognitive functions, such as language.

Results

The UK Biobank data contain information on human handedness and previous studies on structural hemispheric asymmetries in the UK Biobank data have made links between structural asymmetries and handedness, e.g.:

Sha Z, Pepe A, Schijven D, Carrión-Castillo A, Roe JM, Westerhausen R, Joliot M, Fisher SE, Crivello F, Francks C. Handedness and its genetic influences are associated with structural asymmetries of the cerebral cortex in 31,864 individuals. Proc Natl Acad Sci U S A. 2021 Nov 23;118(47):e2113095118. It would be highly interesting and relevant if the authors could see how the findings are linked to handedness.

We thank the reviewer for bringing up this interesting point. We performed a supplementary analysis directed at the association of derived brain asymmetry patterns and handedness in the UK Biobank sample. Specifically, for each of the eight LDA models dedicated to eight CNVs, we quantified LDA expression for each of the 36,000 UK Biobank participants used in this study. This expression is computed as a weighted sum of regional LDA coefficients and participant's regional asymmetries. In the next step, we used a t-test to compare if the LDA expressions differ between left- and right-handed participants. We did not find a significant difference in LDA expression depending on handedness for any of the eight CNVs. We now summarize these findings in the Results section. We also added this analysis as a Supplementary Figure 2.

Results

To supplement the functional profiling using NeuroSynth, we investigated if derived asymmetry patterns are associated with handedness. Although how genetic variants contribute to handedness remains a conundrum (Kavaklioglu et al., 2106), recent studies point to the role of rare variants in handedness (Schijven et al., 2023). Therefore, we probed for the association between the derived asymmetry pattern and handedness in the 36,000 UK Biobank participants. Nevertheless, we did not find any robust association (Sup. Fig. 2).

Supplementary Figure 2

CNV-specific LDA models are not associated with handedness in the UK Biobank sample

We performed an analysis directed at the association of derived brain asymmetry patterns and handedness in the UK Biobank sample. Specifically, for each of the eight LDA models dedicated to eight CNVs, we quantified LDA expression for each of the 36,000 UK Biobank participants used in this study. This expression is computed as a weighted sum of regional LDA coefficients and participant's regional asymmetries. In the next step, we used a t-test to compare if the LDA expressions differ between left- and right-handed participants. We did not find a significant difference in LDA expression depending on handedness for any of the eight CNVs.

Discussion

“Hence, in neurotypical individuals, the planum temporale is highly structurally lateralized, with 65% of all individuals showing larger left planum temporale; and only 11% of individuals showing larger right planum temporale volume”

There is a longstanding discussion on whether the macrostructural asymmetry of the planum is relevant for function or whether microstructural asymmetries are more relevant. These should therefore also be mentioned as they may also be affected by the genetic variation found and the discussion should be a bit more balanced in this regard.

The following paper may be relevant:

Galuske RA, Schlote W, Bratzke H, Singer W. Interhemispheric asymmetries of the modular structure in human temporal cortex. Science. 2000 Sep 15;289(5486):1946-9. doi: 10.1126/science.289.5486.1946
Ocklenburg S, Friedrich P, Fraenz C, Schlüter C, Beste C, Güntürkün O, Genç E. Neurite architecture of the planum temporale predicts neurophysiological processing of auditory speech. Sci Adv. 2018 Jul 11;4(7):eaar6830. doi: 10.1126/sciadv.aar6830

We thank the reviewer for pointing out this very important topic. We agree with the reviewer that microstructural asymmetries might be of particular importance. However, since the study is based on structural MRI, we are not able to investigate microstructure properties. Therefore, the question of micro- vs. macro- structural asymmetries remains beyond the scope of the manuscript. As suggested by the reviewer, we integrated micro- and macro- structural aspects of asymmetry into our Discussion section.

Hence, in neurotypical individuals, the planum temporale is highly structurally lateralized, with 65% of all individuals showing larger left planum temporale; and only 11% of individuals showing larger right planum temporale volume^{47,59} In terms of functional lateralization, there is a possibility that microstructural and not macrostructural features are more important (Ocklenburg et al., 2018). Postmortem studies highlighted the role of intrinsic microcircuitry organization (Galuske et al., 2000). However, the role of micro- and macro- structure is beyond the scope of this article since the used structural MRI signal captures contributions from various structural components within a voxel and is not able to disentangle microstructure properties (Song et al., 2022).

Methods

Statement: “UK Biobank participants gave written, informed consent for the study, which was approved by the Research Ethics Committee.”

Please also include ethics information for participants that were not from the UK Biobank.

We thank the reviewer for highlighting the potential lack of clarity on our side. The information in the first part of the mentioned paragraph referred to non-UK Biobank participants, while the second part of the paragraph referred to UK Biobank participants. We improved the section in order to provide clear information.

Methods:

In the clinically ascertained cohort, signed consent was obtained from all ~~clinical~~ participants or legal representatives prior to inclusion in the study~~—the investigation~~. The current study, which is a secondary data analysis, was approved by the Research Ethics Board ~~HRB~~ (Project 4165) of the Sainte Justine Hospital, Montreal, Canada. UK Biobank participants gave written, informed consent for the study, which was approved by the Research Ethics Committee. The present analyses were conducted under UK Biobank application number 40980. Further information on the consent procedure can be found online (biobank.ctsu.ox.ac.uk/crystal/field.cgi?id=200).

Asymmetry index:

Please give reference and some more info on why $(L - R) / ((L + R) / 2)$ was used and not the more common $(L - R) / (L + R)$.

We thank the reviewer for this question. We followed the definition by some of the most cited manuscripts, namely:

- *Herbert, M. R. et al. Abnormal asymmetry in language association cortex in autism. Ann. Neurol. 52, 588–596 (2002). - 373 citations*
- *Galaburda, A. M., Corsiglia, J., Rosen, G. D. & Sherman, G. F. Planum temporale asymmetry, reappraisal since Geschwind and Levitsky. Neuropsychologia 25, 853–868 (1987). - 90 citations*
- *Shapleske, J., Rossell, S. L., Woodruff, P. W. & David, A. S. The planum temporale: a systematic, quantitative review of its structural, functional and clinical significance. Brain Res. Brain Res. Rev. 29, 26–49 (1999). - 648 citations*
- *Steinmetz, H. Structure, functional and cerebral asymmetry: in vivo morphometry of the planum temporale. Neurosci. Biobehav. Rev. 20, 587–591 (1996). - 261 citations*

In these manuscripts, the denominator is divided by a factor of two. We now include the citations in the Methods section.

REVIEWER COMMENTS

Reviewer #1 (Remarks to the Author):

I would like to thank the authors for their responsiveness and clarifications.

They have fully addressed my comments.

Reviewer #2 (Remarks to the Author):

The authors have improved their manuscript significantly and have implemented my suggestion or argued convincingly why this was needed. I have no further comments and can recommend acceptance.

Reviewer #3 (Remarks to the Author):

This work aims to understand how rare copy number variations (CNVs) relate to brain asymmetry that is considered to underlie some human-defining cognitive functions. Four large CNVs documented for neuropsychiatric disorders were selected and investigated for associations with a total of 65 brain asymmetry measures in an aggregated cohort of 552 CNV carriers and 290 non-carriers. Linear discriminant association (LDA) analysis was utilized to identify multivariate latent variables of asymmetry measures distinguishing between CNV carriers and non-carriers. Bootstrapping was conducted to locate the regional asymmetry patterns most consistently related to CNVs across 100 runs. Additional GWAS and annotation analysis (NeuroSynth, FUMA) were employed for supplemental evidence and biological interpretations. Overall this work is well-motivated. Meanwhile, several methodological aspects of the paper require clarification before the potential significance of the results can be fully appraised.

While CNVs served as the genetic variables of interest, not much information has been provided for the preprocessing and detection of CNVs, not in the cited paper [Ref. 63] either actually. Microarray data (assuming this was the technology used) could be noisy, and it is not clear how QC was conducted on the original Log R ratio data, which could involve outlier removal, normalization, GC content correction, etc.

In the Method section, it was briefly mentioned that PennCNV and QuantiSNP were used to detect CNVs. Was it the case that either PennCNV or QuantiSNP was used at different sites, or both were used so that only CNVs called by both tools were considered validated for further analysis? While in general less false positive calls for large CNVs, having validations is always a safeguard. Another question relates to how the detected CNVs were recognized as one of the four large structural variations. The CNV calls yielded by PennCNV or QuantiSNP in general have varying boundaries and sizes. What was the criterion for flagging a CNV call as one of the large CNVs of interest? Would be helpful to provide some statistics, such as the range of the CNV left and right boundaries, as well as the histograms of their sizes, to help evaluate the integrity of CNVs.

Another concern is site effect and data harmonization. It may need some clarifications on how the carriers of each CNV (deletion and insertion respectively) break down across sites. The related question is that, if all the CNV carriers/non-carriers were from the same site which did not contribute any non-carrier/carrier samples, it would be a bit challenging to separate and correct the site effect in the imaging data, given that it could be in collinearity with the carrier vs. non-carrier effects. This raises the concern about whether the site effect might partly drive the following LDA analysis which was supposed to capture the CNV effects.

Regarding the LDA analysis, a cross-validation would be informative on how generalizable the results are. And for the bootstrapping analysis, I wonder if it might be an option to select important brain asymmetry measures based on normalized LDA weights or weighted percentage of explained variances, to take into account effect sizes.

Reviewer #3 (Remarks to the Author):

This work aims to understand how rare copy number variations (CNVs) relate to brain asymmetry that is considered to underlie some human-defining cognitive functions. Four large CNVs documented for neuropsychiatric disorders were selected and investigated for associations with a total of 65 brain asymmetry measures in an aggregated cohort of 552 CNV carriers and 290 non-carriers. Linear discriminant association (LDA) analysis was utilized to identify multivariate latent variables of asymmetry measures distinguishing between CNV carriers and non-carriers. Bootstrapping was conducted to locate the regional asymmetry patterns most consistently related to CNVs across 100 runs. Additional GWAS and annotation analysis (NeuroSynth, FUMA) were employed for supplemental evidence and biological interpretations. Overall this work is well-motivated. Meanwhile, several methodological aspects of the paper require clarification before the potential significance of the results can be fully appraised.

We are grateful to the reviewer for the positive assessment of our work.

While CNVs served as the genetic variables of interest, not much information has been provided for the preprocessing and detection of CNVs, not in the cited paper [Ref. 63] either actually. Microarray data (assuming this was the technology used) could be noisy, and it is not clear how QC was conducted on the original Log R ratio data, which could involve outlier removal, normalization, GC content correction, etc.

We thank the reviewer for highlighting this important point. We have reworked the CNV section in Methods to provide a detailed description of CNV detection. The current version includes updated references or detailed quality control of microarray data.

Methods:

The identification of CNVs using SNP array (GRCh37/hg19) data followed previously published methods (Huguet et al., 2018; Huguet et al., 2021) and consists of the following steps:

- 1) Array quality control was performed based on standard protocols. Using PLINK v1.9 (Chang et al., 2015), we removed SNP variants with a missing rate > 5% as well as SNPs with a Hardy-Weinberg equilibrium exact test p-value < 0.0001. We only considered arrays with call rate \geq 99%, log R ratio SD < 0.35, B allele frequency SD < 0.08, and the absolute value of wave factor < 0.05. Furthermore, all individuals with duplicated data or with discordant phenotypic and genetic information regarding sex were removed.*
- 2) CNVs were called using the pipeline described at <https://github.com/labjacquemont/MIND-GENESPARALLELCNV>. In short, only CNV detected by both PennCNV (Wang et al., 2007) and QuantiSNP (Collela et al., 2007) were used to minimize the number of potential false positives. The resulting CNV calls are available for download from the UK Biobank returned datasets (Return ID: 3104, <https://biobank.ndph.ox.ac.uk/ukb/dset.cgi?id=3104>). All identified CNVs met stringent quality control criteria: confidence score \geq 30 (for at least one of the two detection algorithms), size \geq 50 kb, unambiguous type (deletion or duplication), overlap with segmental duplicates, and HLA regions or centromeric regions <50%. Finally, all carriers of a structural variant \geq 10Mb, a mosaic CNV or a chromosome anomaly (aneuploidy or sexual chromosome anomaly) were removed.*

- 3) *Recurrent CNVs at the 4 selected genomic loci were defined based on the following criteria:*
i) *The CNV shows reciprocal overlap > 40% with one of the 4 loci: 16p11.2 proximal (BP4-5, 29.6-30.2MB), 1q21.1 distal (Class I, 146.4-147.5MB & II, 145.3-147.5MB), 22q11.2 proximal (BPA-D, 18.8-21.7MB) and 15q11.2 (BP1-2, 22.8–23.0MB).* ii) *CNV includes all of the coding genes within one of the 4 unique genomic loci. As a result, recurrent CNVs at a given loci are 100% identical with respect to coding regions. These CNVs were visually inspected by at least two researchers to confirm these criteria.*

Huguet, G. et al. Measuring and Estimating the Effect Sizes of Copy Number Variants on General Intelligence in Community-Based Samples. JAMA Psychiatry 75, 447–457 (2018).

Huguet, G. et al. Genome-wide analysis of gene dosage in 24,092 individuals estimates that 10,000 genes modulate cognitive ability. Mol Psychiatry 26, 2663–2676 (2021).

In the Method section, it was briefly mentioned that PennCNV and QuantiSNP were used to detect CNVs. Was it the case that either PennCNV or QuantiSNP was used at different sites, or both were used so that only CNVs called by both tools were considered validated for further analysis? While in general less false positive calls for large CNVs, having validations is always a safeguard. Another question relates to how the detected CNVs were recognized as one of the four large structural variations. The CNV calls yielded by PennCNV or QuantiSNP in general have varying boundaries and sizes. What was the criterion for flagging a CNV call as one of the large CNVs of interest? Would be helpful to provide some statistics, such as the range of the CNV left and right boundaries, as well as the histograms of their sizes, to help evaluate the integrity of CNVs.

We thank the reviewer for bringing up the need for more clarity on our side. In short, the CNV detection was performed using both PennCNV and QuantiSNP to minimize the number of potential false discoveries. Specifically, only CNVs called by both PennCNV and QuantiSNP were considered in our analysis (refer to the answer to question 1 for full details).

All 4 large recurrent CNVs were identified based on the following criteria: all genes in the unique (not segmental duplication) genomic segments flanked segmental duplications had to be included in the CNV. The following breakpoints according to the reference genome GRCh37/hg19 were used: 16p11.2 proximal (BP4-5, 29.6-30.2MB), 1q21.1 distal (Class I, 146.4-147.5MB & II, 145.3-147.5MB), 22q11.2 proximal (BPA-D, 18.8-21.7MB) and 15q11.2 (BP1-2, 22.8–23.0MB). In addition, all recurrent CNVs at these 4 loci were visually inspected.

The recurrent CNVs considered in this study have the exact same coding gene content. Their unique genomic region is 100% identical. Their exact boundaries are, however unclear and lay somewhere within the segmental duplications flanking each region. These segmental duplication regions can not be mapped SNP arrays or any other genomic technology except for long-read sequencing.. We agree with the reviewer that whether differences in CNV boundaries within these segmental duplications are associated with phenotypic effects is an important and unresolved question in the literature that would require long-read sequencing methods to investigate.

We have now extended the description of CNV detection in the Methods section with detailed descriptions of the used methodology (refer to the answer to question 1 for full details).

Another concern is site effect and data harmonization. It may need some clarifications on how the carriers of each CNV (deletion and insertion respectively) break down across sites. The related question is that, if all the CNV carriers/non-carriers were from the same site which did not contribute any non-carrier/carrier samples, it would be a bit challenging

to separate and correct the site effect in the imaging data, given that it could be in collinearity with the carrier vs. non-carrier effects. This raises the concern about whether the site effect might partly drive the following LDA analysis, which was supposed to capture the CNV effects.

We thank the reviewer for highlighting this important point. We confirm that all sites contributed CNV carriers and non-carriers, so there are no instances where the CNV effect was co-linear with site effect. Nevertheless, as a data-cleaning step, all derived regional brain volumes were adjusted for variation that can be explained by the scanning site (see Methods). Overall, multisite data aggregation, which is increasingly used in recent studies, improves generalizability (i.e., findings are supported by data collected at different sites around the world), and there are multiple sensitivity analyses that are being used to ensure that site effects are not introducing biases.

To provide more elements to our reviewer's response to the suggestion, we conducted two sensitivity analyses. Namely, we investigated the association between the CNV-specific scores and recruitment site using one-way ANOVA, and leave-one-site-out testing. The outcomes of these analyses are now presented in Supplementary Figure 4. We did not observe a significant difference in LDA scores depending on the scanning site for any of the eight CNVs in any of the two analyses.

Methods:

As a data-cleaning step, all derived regional brain volumes were adjusted for variation that can be explained by the scanning site (Sup. Fig. 4).

Supplementary Figure 4

Low-dimensional summaries of brain-imaging features do not depend on the recruitment site for any of the eight CNVs.

We conducted two sensitivity analyses dedicated to probe the effect of the recruitment site on the derived LDA scores separately for each CNV. A) Leave-one-site-out analysis. For a single CNV (here 1q21.2 deletion), we removed the LDA scores of subjects recruited at a given site (i.e., Cardiff). This reduced set of LDA scores is depicted using the raincloud plot, which combines a scatter plot, a box plot (whiskers equal to 1.5 times the interquartile range), and a violin plot. We then compared the remaining LDA scores with the original LDA scores of all subjects carrying the CNV using a two-sample t-test. Such procedure was repeated for every site, and minimal p-value across sites was used to quantify the effects of the recruitment site on the LDA scores for a given CNV. B) Results of sensitivity analyses. The second sensitivity analysis consisted of employing one-way ANOVA separately for each CNV. The resulting p-values were corrected from multiple comparisons using the FDR procedure across all CNVs separately for each analysis. As displayed in the heatmap, we did not observe a significant relationship between LDA scores and a site for any of the CNVs. Collectively, our set of sensitivity analyses demonstrated that the recruitment site did not drive the obtained LDA solutions.

Regarding the LDA analysis, a cross-validation would be informative on how generalizable the results are.

We thank the reviewer for the interesting remark. Regarding the suggestion for cross-validation, we want to clarify that the primary objective of our LDA analysis was to delve deeply into the patterns and insights within the provided dataset. The emphasis was on understanding the inherent structure and relationships present in the specific data under investigation, rather than building a raw predictive tool such as for application in clinical settings (Bzdok & Ioannidis, 2019; Bzdok et al., 2018). While we acknowledge the importance of cross-validation for assessing the generalizability of results across diverse datasets, our analytical approach for this study was intentionally focused on the unique characteristics of the dataset at hand. We aimed to provide a comprehensive exploration and interpretation of the data available to us. Based on these assumptions, we opted for bootstrapping as a form of model validation (Steyerberg 2019). In our manuscript, we focus on the interpretation of the effects validated by bootstrapping (see Figure below)

Bootstrap significance test. As an acid test for robust brain region effects, we embraced a bootstrapping resampling strategy for the LDA models separately for all eight CNV classes. In the

first phase, a randomly perturbed version of the dataset was created by sampling a subject cohort with the same sample size with replacement. We repeated the bootstrap resampling procedure with 100 iterations. In so doing, we obtained different realizations of the entire analysis workflow and ensuing LDA model estimates. Statistically salient coefficients had the distribution of 100 LDA coefficients significantly different from 0. Specifically, they were robustly different from zero if their two-sided confidence interval according to the 2.5/97.5% bootstrap-derived distribution did not include zero coefficient value, indicating the absence of an effect. The figure depicts the example of LDA coefficients in four selected regions across 1000 bootstrap iterations for the LDA model dedicated to 16p11.2 deletion. In this example, only coefficients for *planum temporale* passed the bootstrap significance test.

Bzdok, D., Altman, N. & Krzywinski, M. *Statistics versus machine learning*. *Nature Methods* 15, 233–234 (2018).

Bzdok, D. & Ioannidis, J. P. A. *Exploration, Inference, and Prediction in Neuroscience and Biomedicine*. *Trends in Neurosciences* 42, 251–262 (2019).

Steyerberg, E. W. *Clinical Prediction Models: A Practical Approach to Development, Validation, and Updating*. (Springer International Publishing, 2019). doi:10.1007/978-3-030-16399-0.

And for the bootstrapping analysis, I wonder if it might be an option to select important brain asymmetry measures based on normalized LDA weights or weighted percentage of explained variances, to take into account effect sizes.

We thank the reviewer for highlighting the lack of clarity. In fact, important brain asymmetry measures were based on normalized LDA weights, as suggested by the reviewer. Each LDA model was based on asymmetry measurements z-scored across all relevant participants. Normalized LDA coefficients provide a standardized measure of each feature's contribution to the discriminant function. The normalization process typically involves dividing each LDA weight by the standard deviation of the corresponding feature. However, since the standard deviation of the original feature was equal to 1, the normalization step did not change LDA coefficients. We now stress the z-scoring step in the Methods section.

Methods:

In our study, we brought to bear LDA models to separate between CNV carriers and controls based on asymmetry measures derived from brain atlas region volumes. **These asymmetry measures were z-scored across relevant participants.** Specifically, we derived a single LDA prototype for each CNV type, which yielded eight CNV-specific models.

REVIEWERS' COMMENTS

Reviewer #3 (Remarks to the Author):

I appreciate the authors' efforts to address the comments. The manuscript is significantly improved.

One additional comment: for the leave-one-site-out analysis (supplemental Figure 4), I would recommend also re-running LDA with one site excluded and examining the resulting LDA coefficients of brain asymmetry measures for stability.

Reviewer #3 (Remarks to the Author):

I appreciate the authors' efforts to address the comments. The manuscript is significantly improved.

One additional comment: for the leave-one-site-out analysis (supplemental Figure 4), I would recommend also re-running LDA with one site excluded and examining the resulting LDA coefficients of brain asymmetry measures for stability.

We thank the reviewer for the positive assesment of our work. As suggested, we extended Supplementary Figure 4 to also examine the stability of LDA coefficients. We ran leave-one-site-out sensitivity analysis similar to the one applied to LDA scores. Results show that site did not have a significant effect on LDA coefficients.

Supplementary Figure 5

Low-dimensional summaries of brain-imaging features do not depend on the recruitment site for any of the eight CNVs

We conducted two sensitivity analyses dedicated to probe the effect of the recruitment site on the derived LDA scores separately for each CNV. A) Leave-one-site-out analysis. For a single

CNV (here 1q21.2 deletion), we removed the LDA scores of subjects recruited at a given site (i.e., Cardiff). This reduced set of LDA scores is depicted using the raincloud plot, which combines a scatter plot, a box plot (whiskers equal to 1.5 times the interquartile range), and a violin plot. We then compared the remaining LDA scores with the original LDA scores of all subjects carrying the CNV using a two-sample t-test. Such procedure was repeated for every site, and minimal p-value across sites was used to quantify the effects of the recruitment site on the LDA scores for a given CNV. B) Results of sensitivity analyses. The second sensitivity analysis consisted of employing one-way ANOVA separately for each CNV. Finally, we also repeated leave-one-site-out analysis to examine the stability of LDA coefficients. For each CNV, we removed the asymmetry measurements of subjects recruited at a given site. This reduced set of participants was used to derive LDA coefficients. We then compared the new set of LDA coefficients with the original LDA coefficients of all subjects carrying the CNV using a two-sample t-test. Such procedure was repeated for every site, and minimal p-value across sites was used to quantify the effects of the recruitment site on the LDA coefficients for a given CNV. All resulting p-values were corrected from multiple comparisons using the FDR procedure across all CNVs separately for each analysis. As displayed in the heatmap, we did not observe a significant relationship between LDA scores or coefficients and a site for any of the CNVs. Collectively, our set of sensitivity analyses demonstrated that the recruitment site did not drive the obtained LDA solutions.